# OUT-OF-DOMAIN UNLABELED DATA IMPROVES GENERALIZATION

**Amir Hossein Saberi** [‡*]          Amir Najafi [†]          Alireza Heidari [†]
Mohammad Hosein Movasaghinia [†]          Seyed Abolfazl Motahari [†]          Babak H. Khalaj[‡§] [*]

[*] Department of Electrical Engineering,
[†] Department of Computer Engineering,
[‡]Sharif Center for Information Systems and Data Science,
[§]Sharif Institute for Convergence Science & Technology,
Sharif University of Technology, Tehran, Iran

## ABSTRACT

We propose a novel framework for incorporating unlabeled data into semi-supervised classification problems, where scenarios involving the minimization of either i) adversarially robust or ii) non-robust loss functions have been considered. Notably, we allow the unlabeled samples to deviate slightly (in total variation sense) from the in-domain distribution. The core idea behind our framework is to combine Distributionally Robust Optimization (DRO) with self-supervised training. As a result, we also leverage efficient polynomial-time algorithms for the training stage. From a theoretical standpoint, we apply our framework on the classification problem of a mixture of two Gaussians in $\mathbb{R}^d$, where in addition to the $m$ independent and labeled samples from the true distribution, a set of $n$ (usually with $n \gg m$) out of domain and unlabeled samples are given as well. Using only the labeled data, it is known that the generalization error can be bounded by $\propto (d/m)^{1/2}$. However, using our method on both isotropic and non-isotropic Gaussian mixture models, one can derive a new set of analytically explicit and non-asymptotic bounds which show substantial improvement on the generalization error compared to ERM. Our results underscore two significant insights: 1) out-of-domain samples, even when unlabeled, can be harnessed to narrow the generalization gap, provided that the true data distribution adheres to a form of the "cluster assumption", and 2) the semi-supervised learning paradigm can be regarded as a special case of our framework when there are no distributional shifts. We validate our claims through experiments conducted on a variety of synthetic and real-world datasets.

## 1 INTRODUCTION

Semi-supervised learning has long been a focal point in the machine learning literature, primarily due to the cost-effectiveness of utilizing unlabeled data compared to labeled counterparts. However, unlabeled data in various domains, such as medicine, genetics, imaging, and audio processing, often originates from diverse sources and technologies, leading to distributional differences between labeled and unlabeled samples. Concurrently, the development of robust classifiers against adversarial attacks has emerged as a vibrant research area, driven by the rise of large-scale neural networks (Goodfellow et al., 2014; Biggio & Roli, 2018). While the primary objective of these methods is to reduce model sensitivity to minor adversarial perturbations, recent observations suggest that enhancing adversarial robustness may also improve the utilization of unlabeled samples (Najafi et al., 2019; Miyato et al., 2018).

This paper aims to demonstrate the efficacy of incorporating out-of-domain unlabeled samples to decrease the reliance on labeled in-domain data. To achieve this, we propose a novel framework inspired by a fusion of concepts from adversarial robustness and self-training. Specifically, we introduce a unique constraint to the conventional Empirical Risk Minimization (ERM) procedure, focusing exclusively on the unlabeled part of the dataset. Our theoretical and experimental analyses

---

[*]Corresponding author: `motahari@sharif.edu`.

show that the inclusion of unlabeled data reduces the generalization gap for both robust and non-robust loss functions. Importantly, our alternative optimization criteria are computationally efficient and can be solved in polynomial time. We have implemented and validated the effectiveness of our method on various synthetic and real-world datasets.

From a theoretical standpoint, akin to prior research (Schmidt et al., 2018; Carmon et al., 2019; Zhai et al., 2019; Alayrac et al., 2019), we also address the binary classification problem involving two Gaussian models in $\mathbb{R}^d$. This problem has been the center of attention in several recent works on theoretical analysis of both semi-supervised and/or adversarially robust learning paradigms. Despite several recent theoretical investigations, the precise trade-off between the sizes of labeled ($m$) and unlabeled ($n$) data, even in this specific case, remains incomplete. A number of works have bounded the labeled sample complexity under the assumption of an asymptotically large $n$ (Kumar et al., 2020), while another series of papers have analyzed this task from a completely unsupervised viewpoint. We endeavor to fill this gap by providing the first empirical trade-off between $m$ and $n$, even when unlabeled data originates from a slightly perturbed distribution. We derive explicit bounds for both robust and non-robust losses of linear classifiers in this scenario. Our results show that as long as $n \geq \Omega\left(m^2/d\right)$, our proposed algorithm surpasses traditional techniques that solely rely on labeled data. We also consider the more general case of non-isotropic Gaussian models, as explored in previous studies.

The remainder of this paper is structured as follows: Section 1.1 provides an overview of related works in distributionally robust optimization and semi-supervised learning. Section 1.3 introduces our notation and definitions. In Section 1.2, we discuss the contributions made by our work. In Section 3, we present our novel method, followed by a theoretical analysis in Section 4. Section 5 showcases our experimental validations, further supporting our theoretical findings. Finally, we draw conclusions in Section 6.

## 1.1 PRIOR WORKS

One of the challenges in adversarially robust learning is the substantial difficulty in increasing the *robust* accuracy compared to achieving high accuracy in non-robust scenarios (Carlini & Wagner, 2017). A study by Schmidt et al. (2018) posited that this challenge arises from the larger sample complexity associated with learning robust classifiers in general. Specifically, they presented a simple model where a good classifier with high standard (non-robust) accuracy can be achieved using only a single sample, while a significantly larger training set is needed to attain a classifier with high robust accuracy. Recent works (Carmon et al., 2019; Zhai et al., 2019; Alayrac et al., 2019) demonstrated that the gap in sample complexity between robust and standard learning, as outlined by Schmidt et al. (2018) in the context of a two-component Gaussian mixture model, can be bridged with the inclusion of unlabeled samples. Essentially, unlabeled samples can be harnessed to mitigate classification errors even when test samples are perturbed by an adversary. Another study by Najafi et al. (2019) achieved a similar result using a different definition of adversarial robustness and a more comprehensive data generation model. Their approach involves the use of 'self-training' to assign soft/hard labels to unlabeled data, contrasting our approach, where unlabeled data is exclusively utilized to constrain the set of classifiers, aiming to avoid crowded regions. While DRO serves as a tool in our approach, it is not necessarily the primary objective. In Deng et al. (2021), authors showed that in the setting of Schmidt et al. (2018), out-of-domain unlabeled samples improve adversarial robustness.

Theoretical analysis of Semi-Supervised Learning (SSL) under the so-called *cluster assumption* has been a long-studied task (Rigollet, 2007). However, beyond Najafi et al. (2019), several recent methods leveraging DRO for semi-supervised learning have emerged (Blanchet & Kang, 2020; Frogner et al., 2021). Notably, Frogner et al. (2021) shares similarities with Najafi et al. (2019); however, instead of assigning artificial labels to unlabeled samples, Frogner et al. (2021) employs them to delimit the ambiguity set and enhance understanding of the marginals. Our work primarily focuses on the *robustness* aspect of the problem rather than advancing the general SSL paradigm.

Defense mechanisms against adversarial attacks usually consider two types of adversaries: i) point-wise attacks similar to Miyato et al. (2018); Nguyen et al. (2015); Szegedy et al. (2013), and ii) distributional attacks (Staib & Jegelka, 2017; Shafieezadeh Abadeh et al., 2015; Mohajerin Esfahani & Kuhn, 2018), where in the case of the latter adversary can change the distribution of data up to a predefined budget. It has been shown that Distributionally Robust Learning (DRL) achieves a

superior robustness compared to point-wise methods (Staib & Jegelka, 2017). Namkoong & Duchi (2017) utilized DRL in order to achieve a balance between the bias and variance of classifier's error, leading to faster rates of convergence compared to empirical risk minimization even in the *non-robust* case. In DRL, the learner typically aims to minimize the loss while allowing the data distribution to vary within an uncertainty neighborhood. The central idea used by Namkoong & Duchi (2017) was to regulate the diameter of this uncertainty neighborhood based on the number of samples. Gao (2022) achieved similar results in DRL while utilizing the *Wasserstein* metric to define the perturbation budget for data distribution. Based on the above arguments, we have also utilized DRL is the main tool in developing our proposed framework.

## 1.2 MAIN CONTRIBUTIONS

We introduce a novel integration of DRO and Semi-Supervised Learning (SSL), leveraging out-of-domain unlabeled samples to enhance the generalization bound of learning problem. Specifically, we theoretically analyze our method in the setting where samples are generated from a Gaussian mixture model with two components, which is a common assumption in several theoretical analyses in this field. For example, a simpler format, when two Gaussians are isotropic and well-separated, is the sole focus of many papers such as Schmidt et al. (2018); Carmon et al. (2019); Alayrac et al. (2019).Some of our notable contributions and improvements over recent works in the field include:

(i) In Theorem 4.1, we present a non-asymptotic bound for adversarially robust learning, leveraging both labeled and unlabeled samples jointly. This result builds upon the work of Carmon et al. (2019) and Alayrac et al. (2019), which focused on the effectiveness of unlabeled samples when a single labeled sample is sufficient for linear classification of a non-robust classifier. However, these studies do not provide insights into the necessary number of unlabeled samples when multiple labeled samples are involved, particularly in scenarios where the underlying distribution exhibits limited separation between the two classes. Our theoretical bounds address and fill this crucial gap.

(ii) Theorem 4.2 introduces a novel non-asymptotic bound for integrating labeled and unlabeled samples in SSL. To underscore the significance of our findings, consider the following example: In the realizable setting, where positive and negative samples can be completely separated by a hyperplane in $\mathbb{R}^d$, the sample complexity of supervised learning for a linear binary classifier is known to be $\mathcal{O}(d/\epsilon)$ Mohri et al. (2018).However, in the non-realizable setting, this complexity escalates to $\mathcal{O}(d/\epsilon^2)$ Mohri et al. (2018).A pivotal question in learning theory revolves around how to approach the sample complexity of $\mathcal{O}(d/\epsilon)$ in the non-realizable setting. Insights provided by Namkoong & Duchi (2017) delve into this inquiry. Notably, even with the awareness that the underlying distribution is a Gaussian mixture, the optimal sample complexity, as per Ashtiani et al. (2018), still exceeds $\mathcal{O}(d/\epsilon^2)$. Our work demonstrates that in scenarios where the underlying distribution is a Gaussian mixture and we possess $m = \mathcal{O}(d/\epsilon)$ labeled samples, coupled with $n = \mathcal{O}\left(\frac{d}{\epsilon^6}\right)$ unlabeled samples (without knowledge of the underlying distribution), one can achieve an error rate lower than or equal to the case of having access to $\mathcal{O}(d/\epsilon^2)$ labeled samples.

(iii) We formalize the incorporation of *out-of-domain* unlabeled samples into the generalization bounds of both robust and non-robust classifiers in Theorems 4.1, 4.2 and 4.4. We contend that this represents a novel contribution to the field, with its closest counterpart being Deng et al. (2021). Notably, Deng et al. (2021) addresses a scenario where the underlying distribution is a isotropic Gaussian mixture with well-separated Gaussian components, while the separation of components is not a prerequisite for our results.

## 1.3 NOTATION AND DEFINITIONS

Let us denote the feature space by $\mathcal{X} \subseteq \mathbb{R}^d$, and assume $\mathcal{H}$ as a class of binary classifiers parameterized by the parameter set $\Theta$: for each $\theta \in \Theta$, we have a classifier $h_\theta \in \mathcal{H}$ where $h_\theta : \mathcal{X} \to \{-1, 1\}$. Assume a positive function $\ell : (\mathcal{X} \times \{-1, 1\} \times \Theta) \to \mathbb{R}_{\geq 0}$ as the loss function. Also, let $P$ be the unknown data distribution over $\mathcal{X} \times \{-1, 1\}$, and $S = \{(\boldsymbol{X}_i, y_i)\}_{i=1}^m$ for $m \in \mathbb{N}$ be a set of i.i.d. samples drawn from $P$. Then, for all $\theta \in \Theta$ the true risk $R$ and the empirical risk $\hat{R}$ of a classifier w.r.t. $P$ can be defined as follows:

$$R(\theta, P) = \mathbb{E}_P[\ell(\boldsymbol{X}, y; \theta)] \quad , \quad R(\theta, \hat{P}_S^m) = \mathbb{E}_{\hat{P}_S^m}[\ell(\boldsymbol{X}, y; \theta)] \triangleq \frac{1}{m}\sum_{i=1}^m \ell(\boldsymbol{X}_i, y_i; \theta), \quad (1)$$

where $\hat{P}_S^m$ denotes an empirical estimate of $P$ based on the $m$ samples in $S$. We also need a way to measure the distance between various distributions that are supported over $\mathcal{X}$. A well-known candidate for this goal is the *Wasserstein* distance (Definition A.1). Subsequently, we also define a *Wasserstein ball* in Definition A.2 in order to effectively constrain a set of probability measures. It should be noted that throughout this paper, the Wasserstein distance between any two distributions supported over $\mathcal{X} \times \{\pm 1\}$ is defined as the distance between their respective marginals on $\mathcal{X}$.

The ultimate goal of classical learning is to find the parameter $\theta^* \in \Theta$ such that with high probability, $R(\theta^*)$ is sufficiently close to $\min_\theta R(\theta)$. A well-known approach to achieve this goal is Empirical Risk Minimization (ERM) algorithm, formally defined as follows:

$$\hat{\theta}^{\mathrm{ERM}}(S) \triangleq \underset{\theta \in \Theta}{\arg\min}\ \mathbb{E}_{\hat{P}_S^m}\left[\ell\left(\theta; \boldsymbol{X}, y\right)\right] = \underset{\theta \in \Theta}{\arg\min}\ \frac{1}{m}\sum_{i=1}^m \ell\left(\theta; \boldsymbol{X}_i, y_i\right). \tag{2}$$

A recent variant of ERM, which has gained huge popularity in both theory and practice, is the so-called Distributionally Robust Learning (DRL) which is formulated as follows:

**Definition 1.1** (Distributionally Robust Learning(DRL)). DRL aims at training a classifier which is robust against adversarial attacks on data distribution. In this regard, the *learner* attempts to find a classifier with a small robust risk, denoted as $R^{\mathrm{robust}}(\theta, P)$, which is defined as

$$R_{\epsilon,c}^{\mathrm{robust}}(\theta, P) = \sup_{P' \in \mathcal{B}_\epsilon^c(P)} R(\theta, P'), \tag{3}$$

for all $\theta \in \Theta$ and any $\epsilon \geq 0$. Therefore, DRL solves the following optimization problem:

$$\hat{\theta}_{\epsilon,c}^{\mathrm{DRL}}(S) \triangleq \underset{\theta \in \Theta}{\arg\min}\ R_{\epsilon,c}^{\mathrm{robust}}\left(\theta, \hat{P}_S^m\right). \tag{4}$$

Surprisingly, the sophisticated minimax optimization problem of equation 4 which takes place in a subset of the infinite-dimensional space of probability measures that corresponds to the constraints, can be substantially simplified when is re-written in the dual format:

**Lemma 1.2** (From Blanchet et al. (2019)). *For a sufficiently small $\epsilon > 0$, the minimax optimization problem of equation 4 has the following dual form:*

$$\inf_{\theta \in \Theta} \sup_{P' \in \mathcal{B}_\epsilon^c\left(\hat{P}_S^m\right)} R(\theta, P') = \inf_{\gamma \geq 0}\left\{\gamma\epsilon + \inf_{\theta \in \Theta}\frac{1}{m}\sum_{i=1}^m \sup_{\boldsymbol{Z} \in \mathcal{X}}\ \ell\left(\boldsymbol{Z}, y_i; \theta\right) - \gamma c\left(\boldsymbol{Z}, \boldsymbol{X}_i\right)\right\}, \tag{5}$$

*where $\gamma$ and $\epsilon$ are dual parameters, and there is a bijective and reciprocal relation between the $\epsilon$ and $\gamma^*$, i.e., the optimal value which minimizes the r.h.s.*

As suggested by Sinha et al. (2017), the $\inf_{\gamma \geq 0}$ in the r.h.s. part in the above optimization problem can be removed by fixing a user-defined value for $\gamma$. This also means that if one attempts to find the optimal value for $\theta$, the additive term $\gamma\epsilon$ is ineffective and can be removed as well.

It should be noted that this also fixes an (unknown) value for $\epsilon$. In practice, the appropriate value for $\epsilon$ is not known beforehand and thus can be usually found through a cross-validation stage, while the same procedure can be applied to its dual counterpart, i.e., $\gamma$. In other words, the above-mentioned strategy keeps the generality of the problem intact. For the sake of simplicity in relations, throughout the rest of the paper we work with the dual formulation in equation 5 and let $\gamma$ be a fixed and arbitrary value.

## 2 PROBLEM DEFINITION

At this point, we can formally define our problem. Let $\mathcal{X} \subseteq \mathbb{R}^d$, and assume $P_0$ be an unknown and arbitrary distribution supported on $\mathcal{X} \times \{\pm 1\}$, i.e., $P_0$ produces feature-label pairs. For a valid cost function $c : \mathcal{X}^2 \to \mathbb{R}_{\geq 0}$, let $P_1$ represent a shifted version of $P_0$ such that the marginal distributions of $P_0$ and $P_1$ on $\mathcal{X}$ are shifted with $\mathcal{W}_c\left(P_{0,X}, P_{1,X}\right) = \alpha$ for some $\alpha > 0$. No assumption on $P_1\left(y|\boldsymbol{X}\right)$ is necessary in this work. Here, the subscript $X$ implies the marginal distribution on $\mathcal{X}$. Let us consider the following two sets of samples:

$$S_0 = \left\{\left(\boldsymbol{X}_i, y_i\right)\right\}_{i=1}^m \sim P_0^m \quad , \quad S_1 = \left\{\boldsymbol{X}_i'\right\}_{i=1}^n \sim P_{1,X}^n,$$

where $S_0$ indicates the labeled set and $S_1$ represents the unlabeled out-of-domain data. A classical result from VC-theory states that the generalization gap in learning from only $S_0$ (with high probability) can be bounded as

$$R\left(\hat{\theta}^{\text{ERM}}, P_0\right) \leq \min_{\theta \in \Theta} R\left(\theta, P_0\right) + \mathcal{O}\left(\sqrt{\text{VCdim}\left(\mathcal{H}\right)/m}\right) + \sqrt{\mathcal{O}(1)/m}, \tag{6}$$

where $\text{VCdim}\left(\mathcal{H}\right)$ denotes the VC-dimension of hypothesis class $\mathcal{H}$ (Mohri et al., 2018). This bound can be prohibitively large when $\text{VCdim}\left(\mathcal{H}\right)$ grows uncontrollably, e.g., the case of linear classifiers in very high dimensions ($d \gg 1$).

We aim to propose a general framework that leverages both $S_0$ and $S_1$ concurrently, and outputs (in polynomial time) an estimator, denoted by $\hat{\theta}^{\text{RSS}}$, such that the second term in the r.h.s. of equation 6 would decay faster as one increases both $m$ and $n$. We are specially interested in cases where $n \gg m$. In the next step, we apply our method on a simplified theoretical example in order to give explicit bounds. Similar to Schmidt et al. (2018); Carmon et al. (2019); Zhai et al. (2019); Alayrac et al. (2019), we fully focus the binary classification problem of a high-dimensional Gaussian mixture model with two components using linear classifiers. Mathematically speaking, for some $\sigma_0 \geq 0$ and $\boldsymbol{\mu}_0 \in \mathbb{R}^d$, let $P_0$ be the feature-label joint distribution over $\mathbb{R}^d \times \{-1, 1\}$ as follows:

$$P_0\left(y = 1\right) = \frac{1}{2}, \quad P_0\left(\boldsymbol{X}|y\right) = \mathcal{N}\left(y\boldsymbol{\mu}_0, \sigma_0^2 \boldsymbol{I}\right). \tag{7}$$

Also, suppose a shifted version of $P_0$, denoted by $P_1$ with $P_{1,X} = (1/2) \sum_{u=-1,1} \mathcal{N}\left(u\boldsymbol{\mu}_1, \sigma_1^2 \boldsymbol{I}\right)$, where $\|\boldsymbol{\mu}_0 - \boldsymbol{\mu}_1\| \leq \mathcal{O}\left(\alpha\right)$ and $|\sigma_1 - \sigma_0| \leq \mathcal{O}\left(\alpha\right)$ [1]. Given the two sample sets $S_0$ and $S_1$ in this configuration, the problem is to estimate the optimal linear classifier which achieves the minimum error rate.

## 3 PROPOSED METHOD: ROBUST SELF SUPERVISED (RSS) TRAINING

We propose a solution that combines two generally independent paradigms in machine learning: self-training (Grandvalet & Bengio, 2004; Amini & Gallinari, 2002), and distributionally robust learning in equation 4. The essence of self-training is to use the currently learned model in order to induce artificial labels on the unlabeled data. Thus, for an unlabeled sample $\boldsymbol{X}'_j$ and any given model parameter $\theta \in \Theta$, one can temporarily consider a pseudo label given by $h_\theta\left(\boldsymbol{X}'_j\right)$. In this regard, the proposed solution denoted by $\hat{\theta}^{\text{RSS}} = \hat{\theta}^{\text{RSS}}\left(S_0, S_1\right)$ can be defined as follows:

**Definition 3.1** (Robust Self-Supervised (RSS) Training). The essence of RSS training is to add a penalty term to the robust version of the original ERM formulation, which is solely evaluated from the out-of-domain unlabeled samples in $S_1$. Mathematically speaking, for a cost function $c$ and parameter $\gamma \geq 0$, let us define the *robust loss* $\phi_\gamma : \mathcal{X} \times \{\pm 1\} \times \Theta \to \mathbb{R}$ as

$$\phi_\gamma\left(\boldsymbol{X}, y; \theta\right) \triangleq \sup_{\boldsymbol{Z} \in \mathcal{X}} \ell\left(\boldsymbol{Z}, y; \theta\right) - \gamma c\left(\boldsymbol{Z}, \boldsymbol{X}\right). \tag{8}$$

In this regard, for a given set of parameters $\gamma, \gamma', \lambda \in \mathbb{R}_{\geq 0}$, the proposed RSS estimator is defined as

$$\hat{\theta}^{\text{RSS}} \triangleq \underset{\theta \in \Theta}{\arg\min} \left\{ \frac{1}{m} \sum_{i=1}^{m} \phi_\gamma\left(\boldsymbol{X}_i, y_i; \theta\right) + \frac{\lambda}{n} \sum_{j=1}^{n} \phi_{\gamma'}\left(\boldsymbol{X}'_j, h_\theta\left(\boldsymbol{X}'_j\right); \theta\right) \right\}. \tag{9}$$

The proposed RSS loss in equation 9 comprises of two main terms. The first term attempts to minimize the empirical robust risk over the labeled data in $S_0$, where an adversary can alter the distribution of samples within a Wasserstein radius characterized by $\gamma$. In the proceeding sections, we show that $\gamma$ can become asymptotically large (radius becomes infinitesimally small) as $m \to \infty$ which is similar to Gao (2022). In fact, a small (but non-zero) budget for the adversary can control the generalization. The second term works only on the unlabeled data which are artificially labeled by $h_\theta$. It can be shown that this term regulates the classifier by forcing it to avoid *crowded* areas. The sensitivity of such regularization is controlled by both $\lambda$ and also $\gamma'$.

---

[1] Having a Wasserstein distance of $\alpha$ between two high-dimensional Gaussian distributions implies that both mean vectors $\boldsymbol{\mu}_0, \boldsymbol{\mu}_1$ and variances $\sigma_0, \sigma_1$ are within a fraction of at most $\mathcal{O}\left(\alpha\right)$ from each other.

## 3.1 MODEL OPTIMIZATION: ALGORITHM AND THEORETICAL GUARANTEES

It can be shown that the for a convex loss function $\ell$, convex cost function $c$, and sufficiently large $\gamma$ and $\gamma'$ (i.e., sufficiently small Wasserstein radii), the optimization problem of equation 9 is convex and can be solved up to an arbitrarily high precision in polynomial time. Moreover, if $\ell$ is not convex, e.g., $\mathcal{H}$ is the set of all neural networks, a simple Stochastic Gradient Descent (SGD) algorithm is still guaranteed to reach to at least a local minimum of equation 9. More specifically, equation 9 is a minimax optimization problem and consists of an inner maximization (formulated in equation 8) followed by an outer minimization. As long as the cost function $c$ is strictly convex and $\gamma$ or $\gamma'$ are chosen sufficiently large, the inner maximization problem of equation 8 becomes strictly concave (Najafi et al., 2019; Sinha et al., 2017). This interesting property holds regardless the convexity of $\ell$, which is of paramount importance since $\ell$ is not convex in most practical situations. On the other hand, cost function candidates for $c$ which are considered in this paper are $\|\cdot\|_2$ and $\|\cdot\|_2^2$, which are strictly convex. Hence, equation 8 can be optimally solved in polynomial time.

The outer minimization problem of equation 9 is also differentiable as long as $\ell$ is sufficiently smooth (again, convexity is not needed). This means the gradient of equation 9 exists and can be efficiently computed using the *Envelope Theorem*. Explicit bounds on the maximum number of steps in a simple SGD algorithm (with a mini-batch size of 1) in order to reach to an $\varepsilon$-neighborhood of the global maximum of equation 8, and a local minimum of equation 9 are given by Sinha et al. (2017). Also, formulating the gradient of minimax loss functions such as equation 9 using the envelope theorem has been carried out, for example, in (Najafi et al., 2019; Sinha et al., 2017). We have also used the same gradient formulation for the numerical optimization of our model parameters in Section 5, where experimental results on real data using neural networks have been illustrated.

In the next section, we derive theoretical guarantees for $\hat{\theta}^{\mathrm{RSS}}$ and show that it leads to improved generalization bounds when $n$ is sufficiently large and $\alpha$ is controlled.

## 4 THEORETICAL GUARANTEES AND GENERALIZATION BOUNDS

In this section, we discuss the theoretical aspects of using the RSS training method, specially for the classification of a two-component Gaussian mixture model using linear classifiers, i.e., $\mathcal{H} \triangleq \left\{ \mathrm{sign}\left(\langle \boldsymbol{\theta}, \cdot \rangle\right) : \mathbb{R}^d \to \{\pm 1\} \mid \boldsymbol{\theta} \in \mathbb{R}^d \right\}$. For the sake of simplicity in results, let us define the loss function $\ell$ as the zero-one loss:

$$\ell\left(\boldsymbol{X}, y; \theta\right) = \mathbf{1}\left(y\langle \theta, \boldsymbol{X} \rangle \geq 0\right). \tag{10}$$

However, extension of the theoretical guarantees in this work to other types of loss functions is straightforward. The following theorem shows that the proposed RSS estimator in 9 can potentially improve the generalization bound in a *robust* learning scenario.

**Theorem 4.1.** *Consider the setup described in Section 2 for the sample generation process (GMM assumption), and the loss function defined in equation 10. Using RSS training with $m$ labeled and $n$ unlabeled samples in $S_0$ and $S_1$, respectively, and for any $\gamma, \delta > 0$, there exist $\lambda$ and $\gamma'$ which can be calculated solely based on input samples such that the following holds with probability at least $1 - \delta$:*

$$\mathbb{E}_{P_0}\left[\phi_\gamma\left(\boldsymbol{X}, y; \hat{\theta}^{\mathrm{RSS}}\right)\right] \leq \min_{\theta \in \Theta} \mathbb{E}_{P_0}\left[\phi_\gamma\left(\boldsymbol{X}, y; \theta\right)\right] \tag{11}$$

$$+ \mathcal{O}\left(\gamma\sqrt{\frac{2d}{m}\left(\alpha\left(\|\boldsymbol{\mu}_0\|_2^2 + \sigma_0^2\right) + \sqrt{\frac{2d}{2n+m}} + \sqrt{\frac{2\log\left(1/\delta\right)}{2n+m}}\right)} + \sqrt{\frac{2\log\left(1/\delta\right)}{m}}\right).$$

The proof, as well as how to calculate $\lambda$ and $\gamma'$ can be found in Appendix B. Theorem 4.1 presents a generalization bound for the proposed estimator when one considers the robust loss under an adversarial budget, which is characterized by $\gamma$. Larger values of $\gamma$ correspond to smaller Wasserstein radii for the distributional adversary of equation 3. The residual term in the r.h.s. of equation 11 converges to zero with a faster rate compared to that of equation 6, given $n$ is sufficiently large and $\alpha$ is sufficiently small. We derive explicit conditions regarding this event in Corollary 4.3. Before that, let us show that for fixed $m$, as one increases the number of unlabeled samples $n$, the *non-robust excess risk* of the RSS-trained classifier decreases as well:

**Theorem 4.2.** *Consider the setting described in Theorem 4.1. Then, the estimator $\hat{\theta}^{\mathrm{RSS}}$ of equation 9 using respectively $m$ labeled and $n$ unlabeled samples, along with specific values of $\gamma, \gamma'$, and $\lambda$ which can be calculated solely from the input samples, satisfies the following non-robust generalization bound with probability at least $1 - \delta$:*

$$R\left(\hat{\theta}^{\mathrm{RSS}}, P\right) - \min_{\theta \in \Theta} R\left(\theta, P\right) \tag{12}$$

$$\leq \mathcal{O}\left( \frac{e^{\frac{-\|\boldsymbol{\mu}_0\|_2^2}{4\sigma_0^2}}}{\sqrt{2\sigma_0\sqrt{2\pi}}} \left( \left(\|\boldsymbol{\mu}_1\|_2^2 + \sigma_1^2\right) \frac{2d\alpha}{m} + \frac{4d}{m}\sqrt{\frac{2d + 2\log\frac{1}{\delta}}{2n+m}} \right)^{1/4} + \sqrt{\frac{2\log\frac{1}{\delta}}{m}} \right).$$

Again, the proof and the procedure for calculating $\gamma, \gamma'$, and $\lambda$ are discussed in Appendix B.

Based on the previous results, the following corollary showcases a number of surprising non-asymptotic conditions under which our generalization bound becomes superior to conventional approaches.

**Corollary 4.3.** *Consider the setting described in Theorem 4.2. Then, $\hat{\theta}^{\mathrm{RSS}}$ of equation 9 with $m$ labeled and $n$ unlabeled samples has an advantage over the traditional ERM, if:*

$$\alpha \leq \mathcal{O}\left(d/m\right) \quad , \quad n \geq \Omega\left(m^2/d\right). \tag{13}$$

*Also, the following conditions are sufficient to make the minimum required $m$ (for a given error bound) independent of the dimension $d$:*

$$\alpha \leq \mathcal{O}\left(d^{-1}\right) \quad , \quad n \geq \Omega\left(d^3\right). \tag{14}$$

Proof is given in Appendix. Finally, Theorem 4.2 also implies that if unlabeled samples are drawn from the same distribution as that of the labeled ones, i.e., $\alpha = 0$, then the excess risk of RSS-training satisfies the following inequality with probability at least $1 - \delta$:

$$R\left(\hat{\theta}^{\mathrm{RSS}}, P\right) - \min_{\theta \in \Theta} R\left(\theta, P\right) \leq \mathcal{O}\left( \left(\frac{d^3\log 1/\delta}{m^2\left(2n+m\right)}\right)^{1/8} + \sqrt{\frac{\log 1/\delta}{m}} \right), \tag{15}$$

which again shows the previously-mentioned improvements when all samples are in-domain.

The assumption of an *isotropic* GMM with two components has been already studied in the literature (see Section 1). Next, we present a more general case of Theorem 4.2 where each Gaussian component can have a non-diagonal covariance matrix. Mathematically speaking, suppose that $P_0$ and $P_1$ are defined as follows:

$$P_0\left(y=1\right) = 1/2 \quad , \quad P_0\left(\boldsymbol{X}|y\right) = \mathcal{N}\left(y\boldsymbol{\mu_0}, \Sigma_0\right),$$

$$P_{1,X} = \frac{1}{2}\mathcal{N}\left(\boldsymbol{\mu_1}, \Sigma_1\right) + \frac{1}{2}\mathcal{N}\left(-\boldsymbol{\mu_1}, \Sigma_1\right), \tag{16}$$

where $\|\boldsymbol{\mu}_1 - \boldsymbol{\mu}_0\| \leq \mathcal{O}\left(\alpha\right), \|\Sigma_1 - \Sigma_0\|_2 \leq \mathcal{O}\left(\alpha\right)$ and $\|\boldsymbol{\mu}_1\|_2 \geq \beta\lambda_{\max}\left(\Sigma_1\right)$. Assume a set of $m$ labeled samples $S_0 \sim P_0^m$, and a set of $n$ unlabeled samples $S_1 \sim P_{1,X}^n$.

**Theorem 4.4** (Generalization Bound for General Gaussian Mixture Models). *Consider the setting described in equation 16. Using algorithm in equation 9 with $m$ labeled and $n$ unlabeled samples, there exists a set of parameters $\gamma, \gamma', \lambda$ for which the following holds with probability at least $1 - \delta$:*

$$R\left(\hat{\theta}^{\mathrm{RSS}}, P\right) - \min_{\theta \in \Theta} R\left(\theta, P\right) \leq \tag{17}$$

$$\mathcal{O}\left( e^{\vartheta^2} \left( \sqrt{\frac{\|\boldsymbol{\mu}_1\|_2^2 + \mathrm{Tr}\left(\Sigma_1\right)}{m}} \left( C\alpha + \sqrt{\frac{\log\frac{1}{\delta}}{2n+m}} \right) \frac{d\kappa_1\kappa_1'}{\Delta\left(\Sigma\right)} \right)^{1/2} + \sqrt{\frac{\log 1/\delta}{m}} \right),$$

*where*

$$\vartheta = |\boldsymbol{\mu}_1\Sigma_1^{-1}\boldsymbol{\mu}_1 - \boldsymbol{\mu}_0\Sigma_0^{-1}\boldsymbol{\mu}_0|, \quad C = \left( \frac{\|\mu_0\|^2 + \lambda_{\min}\left(\Sigma_1\right)\|\mu_0\|_2}{\lambda_{\min}^2} \right),$$

$$\kappa_1 = \frac{\lambda_{\max}\left(\Sigma_1\right)}{\lambda_{\min}\left(\Sigma_1\right)}, \qquad \kappa_1' = \frac{\lambda_{\max}\left(\Sigma_1\right)}{\Delta\left(\Sigma_1\right)},$$

$$\Delta\left(\Sigma_1\right) = \min\left\{\lambda_i\left(\Sigma_1\right) - \lambda_j\left(\Sigma_1\right)\right\}, \quad \forall i,j : \lambda_i\left(\Sigma_1\right) \neq \lambda_j\left(\Sigma_1\right), \tag{18}$$

*and $\lambda_i\left(\Sigma\right)$ is the $i(th)$ eigenvalue of $\Sigma$.*

Table 1: Accuracy of the model trained on labeled datasets of sizes $10$, $20$, $40$, and $10,000$ with varying amounts of unlabeled data from the same distribution with $\alpha = 0$ (**left**), and different distribution with $\alpha = 0.5\|\boldsymbol{\mu}_0\|_2$ (**right**).

| Same distribution | | | | Different distribution | | | |
|---|---|---|---|---|---|---|---|
| Labeled size | Acc | Unlabeled size | Acc | Labeled size | Acc | Unlabeled size | Acc |
| 10 | 0.59 | 10 | 0.63 | 10 | 0.59 | 10 | 0.61 |
| | | 100 | 0.66 | | | 100 | 0.65 |
| | | 1,000 | 0.79 | | | 1,000 | 0.78 |
| | | 10,000 | **0.82** | | | 10,000 | **0.81** |
| 20 | 0.62 | 20 | 0.64 | 20 | 0.62 | 20 | 0.65 |
| | | 200 | 0.69 | | | 200 | 0.65 |
| | | 2,000 | 0.80 | | | 2,000 | 0.79 |
| | | 10,000 | **0.82** | | | 10,000 | **0.80** |
| 40 | 0.65 | 40 | 0.65 | 40 | 0.65 | 40 | 0.65 |
| | | 400 | 0.71 | | | 400 | 0.73 |
| | | 4,000 | 0.81 | | | 4,000 | 0.78 |
| | | 10,000 | **0.82** | | | 10,000 | **0.80** |
| 10,000 | **0.83** | - | - | 10,000 | **0.83** | - | - |

Proof can be found in Appendix. One important difference to note between Theorem 4.4 and Theorem 4.2 is the choice of $\gamma'$, which controls the adversarial budget for unlabeled (and out-of-domain) part of the dataset. In the setting of Theorem 4.2, we prefer to choose $\gamma'$ as small as possible. However, in the setting of Theorem 4.4, we consider the eigenvectors and eigenvalues of $\Sigma_1$ and $\Sigma_0$, as well as the direction of $\boldsymbol{\mu}_1$ and $\boldsymbol{\mu}_0$ in order to find the optimal value for the adversarial budget. In fact, there are cases in which selecting a large $\gamma'$ (less freedom for the adversary) may actually be the optimal choice.

## 5 EXPRIMENTAL RESULTS

The effectiveness of the proposed method has been assessed through experimenting on various datasets, including simulated data, and real-world datasets of histopathology images. Each experiment has been divided into two parts: i) cases in which both labeled and unlabeled data are sampled from the same distribution, and ii) the scenarios where the unlabeled data differs in distribution from the labeled ones. First, let us specify the datasets used in our experiments:

1. **Simulated data** consists of binary-labeled data points with a dimension of $d = 200$, generated according to the setting described in Section 2.

2. **NCT-CRC-HE-100K** consists of 100,000 histopathology images of colon tissue (Katherm et al., 2018). The images have dimensions of $224 \times 224$ and were captured at 20x magnification. The dataset is labeled with 9 distinct classes .

3. **PatchCamelyon** is a widely used benchmark dataset for medical image analysis. It consists of a large collection of 327,680 color histopathology images from lymph node, each with dimensions $96 \times 96$. The dataset has binary labels for presence/absence of metastatic tissue.

### 5.1 EXPERIMENT OF SIMULATED DATA

To evaluate the effectiveness of our method on simulated data, we first find the optimal classifier using only labeled samples. Then, we apply our method with a varying number of unlabeled samples. The results (see Table 1) show that our proposed method achieves accuracy improvements comparable to models trained only on labeled samples. Moreover, results indicate that our method is more effective when labeled and unlabeled data come from the same distribution. However, it still demonstrates significant improvement even when the unlabeled samples undergo a distribution shift.

Table 2: Accuracy of the model trained on labeled data from NCT-CRC-HE-100K dataset with varying amounts of unlabeled data from the same distribution (**left**), as well as when unlabeled samples come from a different distribution (PatchCamelyon dataset)(**right**).

| Same distribution | | | | Different distribution | | | |
|---|---|---|---|---|---|---|---|
| Labeled size | Acc | Unlabeled size | Acc | Labeled size | Acc | Unlabeled size | Acc |
| 48 | 0.65 | 200 | 0.71 | 25 | 0.78 | 100 | 0.78 |
| | | 700 | 0.80 | | | 400 | 0.79 |
| | | 2,000 | **0.82** | | | 2,000 | **0.81** |
| 240 | 0.77 | 500 | 0.78 | 50 | 0.82 | 200 | 0.82 |
| | | 1,200 | 0.82 | | | 700 | 0.86 |
| | | 4,000 | **0.83** | | | 3,000 | **0.87** |
| 1040 | 0.83 | 3,000 | 0.87 | 300 | 0.87 | 600 | 0.88 |
| | | 10,000 | 0.89 | | | 2,000 | 0.89 |
| | | 20,000 | **0.91** | | | 8,000 | **0.90** |
| 50,000 | **0.916** | - | - | 32,000 | **0.94** | - | - |

## 5.2 EXPERIMENT OF HISTOPATHOLOGY DATA

The processing pipeline over the real-world dataset of histopathology images is based on using a ResNet50 encoder pre-trained on ImageNet (Deng et al., 2009; He et al., 2016), which extracts and stores $1 \times 1024$ embeddings from input images. Such embeddings are then used to train a deep neural network with four layers of size $2048$ and one output layer for the class id. Also, we have used a LeakyReLU activation function.

Experimental results in this part are shown in Table 2. Under the "same distribution" setting, both labeled and unlabeled data have been taken from the NCT-CRC-HE-100K dataset. On the other hand, "different distributions" setting implies that labeled data comes from the NCT-CRC-HE-100K dataset (labels are either "Normal" or "Tumor"), while the PatchCamelyon dataset was used for the unlabeled data. As a result, the final labeling is binary. The experimental results demonstrate that increasing the number of unlabeled data leads to an improvement in accuracy for both the 'same' and 'different' distribution settings.

## 6 CONCLUSION

In this study, we address the robust and non-robust classification challenges with a limited labeled dataset and a larger collection of unlabeled samples, assuming a slight perturbation in the distribution of unlabeled data. We present the first non-asymptotic tradeoff between labeled ($m$) and unlabeled ($n$) sample sizes when learning a two-component Gaussian mixture model. Our analysis reveals that when $n \geq \Omega\left(m^2/d\right)$, the generalization bound improves compared to using only labeled data, even when unlabeled data points are slightly out-of-domain. We derive sophisticated results for the generalization error in both robust and non-robust scenarios, employing a technique based on optimizing a robust loss and regularization to avoid crowded and dense areas. Our framework integrates tools from self-training, distributionally robust learning, and optimal transport.

Experiments on synthetic and real-world datasets validate our theoretical findings, demonstrating improved classification accuracy, even for non-Gaussian cases, by incorporating out-of-domain unlabeled samples. Our methodology hinges on leveraging such data to enhance robust accuracy and adapting the uncertainty neighborhood radius based on labeled and unlabeled sample quantities to strike a balance between bias and variance in classification error.

For future work, there's room for improving and relaxing the conditions for the utility of unlabeled data. Exploring error lower-bounds and impossibility results presents another intriguing avenue. Additionally, relaxing the constraints on the level of distribution shift for out-of-domain samples could be a promising direction.

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

## A    AUXILIARY DEFINITIONS

A number of our auxiliary definitions are presented in this section:

**Definition A.1** (Wasserstein Distance). Consider two probability distributions $P$ and $Q$ supported on $\mathcal{X}$, and assume cost function $c : \mathcal{X} \times \mathcal{X} \to \mathbb{R}_+$ is a non-negative lower semi-continuous function satisfying $c(\boldsymbol{X}, \boldsymbol{X}) = 0$ for all $\boldsymbol{X} \in \mathcal{X}$. Then, the Wasserstein distance between $P$ and $Q$ w.r.t. $c$, denoted as $\mathcal{W}_c(P, Q)$, is defined as

$$\mathcal{W}_c(P, Q) = \inf_{\mu \in \Gamma(\mathcal{X}^2)} \mathbb{E}_{\boldsymbol{X}, \boldsymbol{X}' \sim \mu} [c(\boldsymbol{X}, \boldsymbol{X}')], \text{ subject to } \mu(\boldsymbol{X}, \cdot) = P, \ \mu(\cdot, \boldsymbol{X}') = Q, \quad (19)$$

where $\Gamma(\mathcal{X}^2)$ denotes the set of all couplings over $\mathcal{X} \times \mathcal{X}$.

**Definition A.2** ($\epsilon$-neighborhood of a Distribution). The $\epsilon$-neighborhood of a distribution $P$ is defined as the set of all distributions that have a Wasserstein distance less than $\epsilon$ from $P$. Mathematically, it can be represented as:

$$\mathcal{B}_\epsilon^c(P) = \{Q : \mathcal{W}_c(P, Q) \leq \epsilon\}. \quad (20)$$

# B    PROOF OF THEOREMS

In this section, we present the proofs for the Theorems and Lemmas from the main manuscript.

*Proof of Theorem 4.1.* With respect to discussions from the main manuscript, we already know that $\hat{\theta}^{\text{RSS}}$ is the result of the following optimization problem:

$$\hat{\theta}^{\text{RSS}} \triangleq \arg\min_{\theta \in \Theta} \left\{ \frac{1}{m} \sum_{i=1}^m \phi_\gamma(\boldsymbol{X}_i, y_i; \theta) + \frac{\lambda}{n} \sum_{j=1}^n \phi_{\gamma'}(\boldsymbol{X}_j', h_\theta(\boldsymbol{X}_j'); \theta) \right\}, \quad (21)$$

where we have $\Theta \triangleq \{\theta \in \mathbb{R}^d, \|\theta\|_2 = 1\}$. If we consider the loss function mentioned in Theorem 4.1, along with the cost functions $c(\boldsymbol{X}, \boldsymbol{X}') = \|\boldsymbol{X} - \boldsymbol{X}'\|_2$ and $c'(\boldsymbol{X}, \boldsymbol{X}') = \|\boldsymbol{X} - \boldsymbol{X}'\|_2^2$, the Slater conditions will be satisfied. Hence, we have strong duality, and therefore, there exists $s \geq 0$ such that $\hat{\theta}^{\text{RSS}}$ is also the result of the following optimization problem:

$$\hat{\theta}^{\text{RSS}} = \arg\min_{\theta : \frac{1}{n} \sum_{j=1}^n \phi_{\gamma'}(\boldsymbol{X}_j', h_\theta(\boldsymbol{X}_j'); \theta) \leq s} \frac{1}{m} \sum_{i=1}^m \phi_\gamma(\boldsymbol{X}_i, y_i; \theta). \quad (22)$$

To prove the theorem, we need to establish an upper bound on the distance between the expected robust loss of $\hat{\theta}^{\text{RSS}}$ and the best possible expected robust loss. To achieve this, we first provide a concentration bound for the maximum distance between the empirical and expected robust loss across all parameters with

$$\frac{1}{n} \sum_{j=1}^n \phi_{\gamma'}(\boldsymbol{X}_j', h_\theta(\boldsymbol{X}_j'); \theta) \leq s, \quad (23)$$

and then, we demonstrate that the parameter $\theta^\star$, which achieves the best possible robust loss, satisfies equation 23. Suppose that the loss function is bounded with $b$, i.e.,

$$l(\boldsymbol{X}, y; \theta) \leq b, \ \forall \theta, \boldsymbol{X}, y,$$
$$\phi_\gamma^c(\boldsymbol{X}, y; \theta) \leq b, \ \forall c, \gamma, \theta, \boldsymbol{X}, y,$$

where the second inequality directly results from the first one. Therefore, $\frac{1}{m} \sum_{i=1}^m \phi_\gamma(\boldsymbol{X}_i, y_i; \theta)$ has the bounded difference property with parameter $b/m$, which results that

$$\Phi(P_0, S_0, \gamma, c) = \sup_{\theta \in \tilde{\Theta}} \left( \frac{1}{m} \sum_{i=1}^m \phi_\gamma(\boldsymbol{X}_i, y_i; \theta) - \mathbb{E}_{P_0}[\phi_\gamma^c(\boldsymbol{X}, y; \theta)] \right), \quad (24)$$

also has the bounded difference property with parameter $b/m$, where $\tilde{\Theta}$ is defined as

$$\tilde{\Theta} \triangleq \left\{ \theta : \frac{1}{n} \sum_{j=1}^n \phi_{\gamma'}(\boldsymbol{X}_j', h_\theta(\boldsymbol{X}_j'); \theta) \leq s, \|\theta\|_2 = 1 \right\}. \quad (25)$$

So, we have the following inequality with probability more than $1 - \delta$:

$$\Phi\left(P_0, S_0, \gamma, c\right) \leq \mathbb{E}\left[\Phi\left(P_0, S_0, \gamma, c\right)\right] + \sqrt{\frac{\log 1/\delta}{m}}, \quad \forall \theta \in \tilde{\Theta}. \tag{26}$$

In the above inequality we can give an upper-bound for the first term in the right hand side of the inequality as follows:

$$\begin{aligned}
\mathbb{E}\left[\Phi\left(P_0, S_0, \gamma, c\right)\right] =& \mathbb{E}_{S_0 \sim P_0^m}\left[\sup_{\theta \in \tilde{\Theta}}\left(\frac{1}{m}\sum_{i=1}^{m}\phi_\gamma\left(\boldsymbol{X}_i, y_i; \theta\right) - \mathbb{E}_{P_0}\left[\phi_\gamma^c\left(\theta; \boldsymbol{X}, y\right)\right]\right)\right] \\
\leq& \mathbb{E}_{S_0, S_0'}\left[\sup_{\theta \in \tilde{\Theta}}\left(\frac{1}{m}\sum_{i=1}^{m}\phi_\gamma\left(\boldsymbol{X}_i, y_i; \theta\right) - \frac{1}{m}\sum_{i=1}^{m}\phi_\gamma\left(\boldsymbol{X}_i', y_i'; \theta\right)\right)\right] \\
\leq& 2\mathbb{E}_{S_0, \epsilon}\left[\sup_{\theta \in \tilde{\Theta}}\frac{1}{m}\sum_{(\boldsymbol{X}_i, y_i) \in S_0}\epsilon_i \phi_\gamma^c\left(\theta; \boldsymbol{X}_i, y_i\right)\right].
\end{aligned} \tag{27}$$

Consider the loss function mentioned in the Theorem 4.1, and the following cost functions $c, c'$

$$c(\boldsymbol{X}, \boldsymbol{X}') = \|\boldsymbol{X} - \boldsymbol{X}'\|_2, \quad c'(\boldsymbol{X}, \boldsymbol{X}') = \|\boldsymbol{X} - \boldsymbol{X}'\|_2^2.$$

With these loss and cost functions we can rewrite $\phi_\gamma^c$ and $\phi_{\gamma'}^{c'}$ as follows:

$$\phi_\gamma^c\left(\boldsymbol{X}, y; \theta\right) = \min\left\{1, \max\left\{0, 1 - \gamma y\langle\theta, \boldsymbol{X}\rangle\right\}\right\}, \tag{28}$$

$$\phi_{\gamma'}^{c'}\left(\boldsymbol{X}, h_\theta\left(\boldsymbol{X}\right); \theta\right) = \max\left\{0, 1 - \gamma'\langle\theta, \boldsymbol{X}\rangle^2\right\}. \tag{29}$$

As we can see $\phi_\gamma^c\left(\boldsymbol{X}, y; \theta\right)$ is a $\gamma$-lipschitz function of $y\langle\theta, \boldsymbol{X}\rangle$. Based on the Talagrand's contraction lemma, we can continue the inequalities in equation 27 as

$$\begin{aligned}
\mathbb{E}\left[\Phi\left(P + 0, S_0, \gamma, c\right)\right] \leq& \mathbb{E}_{S_0, \epsilon}\left[\sup_{\theta \in \tilde{\Theta}}\frac{1}{m}\sum_{(\boldsymbol{X}_i, y_i) \in S_0}\epsilon_i\phi_\gamma^c\left(\boldsymbol{X}_i, y_i; \theta\right)\right] \\
\leq& \gamma\mathbb{E}_{S_0, \epsilon}\left[\sup_{\theta \in \tilde{\Theta}}\frac{1}{m}\sum_{(\boldsymbol{X}_i, y_i) \in S_0}\epsilon_i y_i\langle\theta, \boldsymbol{X}_i\rangle\right] \\
\leq& \gamma\mathbb{E}_{S_0, \epsilon}\left[\inf_{\zeta > 0}\sup_{\theta \in \Theta}\frac{1}{m}\sum_{(\boldsymbol{X}_i, y_i) \in S_0}\epsilon_i\langle\theta, y_i\boldsymbol{X}_i\rangle - \frac{\zeta}{n}\sum_{\boldsymbol{X}_i' \in S_1}\phi_{\gamma'}^{c'}\left(\boldsymbol{X}_i', h_\theta\left(\boldsymbol{X}_i'\right); \theta\right) + \zeta s'\right] \\
\leq& \gamma\mathbb{E}_{S_0, \epsilon}\left[\inf_{\zeta > 0}\sup_{\theta \in \Theta}\frac{1}{m}\sum_{(\boldsymbol{X}_i, y_i) \in S_0}\epsilon_i y_i\langle\theta, \boldsymbol{X}_i\rangle + \frac{\zeta\gamma'}{n}\sum_{\boldsymbol{X}_i' \in S_1}\langle\theta, \boldsymbol{X}_i'\rangle^2 - \zeta(1 - s)\right] \\
=& \gamma\mathbb{E}_{S_0, \epsilon}\left[\inf_{\zeta > 0}\sup_{\theta \in \Theta}\langle\theta, \frac{1}{m}\sum_{\boldsymbol{X}_i \in S_0}\epsilon_i y_i\boldsymbol{X}_i\rangle + \zeta\gamma'\theta^T\left(\frac{1}{n}\sum_{\boldsymbol{X}_i' \in S_1}\boldsymbol{X}_i'\boldsymbol{X}_i'^T\right)\theta - \zeta(1 - s)\right].
\end{aligned} \tag{30}$$

Let us set $\boldsymbol{v} = \frac{1}{m}\sum_{\boldsymbol{X}_i \in S_0}\epsilon_i y_i\boldsymbol{X}_i$ and $\hat{\Sigma} = \left(\frac{1}{n}\sum_{\boldsymbol{X}_i' \in S_1}\boldsymbol{X}_i'\boldsymbol{X}_i'^T\right)$, and then continue the inequalities in equation 30 as

$$\mathbb{E}\left[\Phi\left(P + 0, S_0, \gamma, c\right)\right] \leq \gamma\mathbb{E}_{S_0, \epsilon}\left[\inf_{\zeta > 0}\sup_{\theta \in \Theta}\langle\theta, \boldsymbol{v}\rangle + \zeta\gamma'\theta^T\hat{\Sigma}\theta - \zeta(1 - s)\right]. \tag{31}$$

In order to find a proper upper-bound for the Rademacher complexity, we attempt to solve the inner maximization problem

$$\sup_{\theta \in \Theta}\langle\theta, \boldsymbol{v}\rangle + \zeta\gamma'\theta^T\hat{\Sigma}\theta. \tag{32}$$

We know that the parameter $\theta$ which maximizes the first term $\langle\theta, \boldsymbol{v}\rangle$ is a vector with Euclidean norm of 1 in the direction of $\boldsymbol{v}$. Also, the parameter which maximizes the second term $\theta^T\hat{\Sigma}\theta$ is a vector

with norm 1 in the direction of $\boldsymbol{u}_{\max}$, which is the eigenvector of $\hat{\Sigma}$ that corresponds to the maximum eigenvalue $\lambda_{\max}$. Therefore, the solution to the maximization problem in equation 32 should belong to the span of $\boldsymbol{v}$ and $\boldsymbol{u}_{\max}$, i.e.,

$$
\begin{aligned}
\theta =& \psi_1 \frac{\boldsymbol{v}}{\|\boldsymbol{v}\|_2} + \psi_2 \frac{\boldsymbol{u}_{\max}}{\|\boldsymbol{u}_{\max}\|_2} \\
=& \psi_1 \hat{\boldsymbol{v}} + \psi_2 \hat{\boldsymbol{u}}_{\max},
\end{aligned}
\tag{33}
$$

where:

$$
\begin{aligned}
& \psi_1^2 + \psi_2^2 + 2\psi_1\psi_2\eta = 1, \\
& \eta = \langle \hat{\boldsymbol{v}}, \hat{\boldsymbol{u}}_{\max} \rangle, \\
& \kappa = \sqrt{1 - \langle \hat{\boldsymbol{v}}, \hat{\boldsymbol{u}}_{\max} \rangle^2} \\
& \tau = \hat{\boldsymbol{v}}^T \hat{\Sigma} \hat{\boldsymbol{v}} \\
& s' = \frac{1-s}{\gamma'}.
\end{aligned}
\tag{34}
$$

Due to equation 32, equation 33 and equation 34, we have the following problem:

$$
\sup \psi_1 \|\boldsymbol{v}\|_2 + \psi_2 \eta.
\tag{35}
$$
$$
\text{S.t. } \psi_1^2 + \psi_2^2 + 2\psi_1\psi_2\eta = 1,
$$
$$
\psi_2^2 \lambda_{\max} + \psi_1^2 \tau + 2\psi_1\psi_2\lambda_{\max}\eta \geq s'
$$

The constraints in the above problem can be rewritten as

$$
\begin{aligned}
& \psi_1^2 + \psi_2^2 + 2\psi_1\psi_2\eta = 1, \\
& \lambda_{\max} - \psi_1^2 (\lambda_{\max} - \tau) \geq s'.
\end{aligned}
\tag{36}
$$

In the above equations, if we set $s' = \lambda_{\max}$, then the only $\psi_1$ which satisfies the constraint is $\psi_1 = 0$. Suppose that we use a set of $n$ i.i.d. samples from $\mathbb{P}_{1X}$ (which is different from the $S_1$) to estimate the $\lambda_{\max}$, and name this estimate $\hat{\lambda}_{\max}$. If we set

$$
s' = \frac{1-s}{\gamma'} = \hat{\lambda}_{\max} (1-\alpha) - 3\mathcal{O}\left( \sqrt{\frac{d}{n}} + \sqrt{\frac{\log(\frac{2}{\delta})}{n}} \right),
\tag{37}
$$

form Theorem 6.5 in Wainwright (2019) we have the following relation with probability more than $1 - \delta$:

$$
\lambda_{\max} (1-\alpha) - 5\mathcal{O}\left( \sqrt{\frac{d}{n}} + \sqrt{\frac{\log(\frac{2}{\delta})}{n}} \right) \leq s' \leq \lambda_{\max} (1-\alpha) - \mathcal{O}\left( \sqrt{\frac{d}{n}} + \sqrt{\frac{\log(\frac{2}{\delta})}{n}} \right).
\tag{38}
$$

By weakening the constraint in equation 35, the result of the optimization becomes larger, which means by solving the following optimization problem we have an upper-bound for equation 35:

$$
\sup \psi_1 \|\boldsymbol{v}\|_2 + \psi_2 \eta.
\tag{39}
$$
$$
\text{S.t. } \psi_1^2 + \psi_2^2 + 2\psi_1\psi_2\eta = 1
$$
$$
\psi_1^2 (\lambda_{\max} - \tau) \leq \alpha\lambda_{\max} + 5\mathcal{O}\left( \sqrt{\frac{d}{n}} + \sqrt{\frac{\log(\frac{2}{\delta})}{n}} \right),
$$

where for the Gaussian distribution and the way $\hat{\boldsymbol{v}}$ is built we have:

$$
\lambda_{\max} - \tau = \|\mu_1\|_2^2 \left( 1 - \frac{\sqrt{\|\mu_0\|_2^2 + \sigma_0^2}}{\sqrt{\|\mu_0\|_2^2 + \sigma_0^2 d}} \right) = \chi^2.
\tag{40}
$$

In this regard, the results of optimization problems equation 39 and equation 32 have the following upper-bound:

$$
\frac{\|v\|_2}{\chi} \sqrt{\alpha\lambda_{\max} + 5\mathcal{O}\left( \sqrt{\frac{d}{n}} + \sqrt{\frac{\log(\frac{2}{\delta})}{n}} \right)} = \frac{\|\frac{1}{m}\sum_{\boldsymbol{X}_i \in S_0} \epsilon_i \boldsymbol{X}_i\|_2}{\chi} \sqrt{\alpha\lambda_{\max} + 5\mathcal{O}\left( \sqrt{\frac{d}{n}} + \sqrt{\frac{\log(\frac{2}{\delta})}{n}} \right)}.
\tag{41}
$$

This way, we also have an upper-bound for Rademacher complexity in equation 27 as

$$\mathbb{E}\left[\Phi\left(P+0, S_0, \gamma, c\right)\right] \leq \gamma \frac{\mathbb{E}_{\boldsymbol{\epsilon}, S_0}\left[\|\frac{1}{m}\sum_{\boldsymbol{X}_i \in S_0}\epsilon_i \boldsymbol{X}_i\|_2\right]}{\chi}\sqrt{\alpha\lambda_{\max} + 5\mathcal{O}\left(\sqrt{\frac{d}{n}} + \sqrt{\frac{\log(\frac{2}{\delta})}{n}}\right)}$$

$$\leq \gamma \frac{\sqrt{\mathbb{E}_{\boldsymbol{X}}\left[\|\boldsymbol{X}\|_2^2\right]}}{\chi\sqrt{m}}\sqrt{\alpha\lambda_{\max} + 5\mathcal{O}\left(\sqrt{\frac{d}{n}} + \sqrt{\frac{\log(\frac{2}{\delta})}{n}}\right)}, \tag{42}$$

where for the Gaussian distribution we have:

$$\mathbb{E}\left[\Phi\left(P+0, S_0, \gamma, c\right)\right] \leq \gamma \frac{\sqrt{\|\mu_1\|_2^2 + \sigma_1^2 d}}{\chi\sqrt{m}}\sqrt{\alpha\lambda_{\max} + 5\mathcal{O}\left(\sqrt{\frac{d}{n}} + \sqrt{\frac{\log(\frac{2}{\delta})}{n}}\right)}$$

$$\leq \gamma \sqrt{\frac{1 + \frac{\sigma_1^2 d}{\|\mu_1\|_2^2}}{m\left(1 - \frac{1}{\sqrt{1 + \frac{\sigma_0^2 d}{\|\mu_0\|_2^2}}}\right)}}\sqrt{\alpha\lambda_{\max} + 5\mathcal{O}\left(\sqrt{\frac{d}{n}} + \sqrt{\frac{\log(\frac{2}{\delta})}{n}}\right)}$$

$$\leq \mathcal{O}\left(\gamma\sqrt{\frac{d}{m}\left(\alpha\lambda_{\max} + \sqrt{\frac{d}{n}} + \sqrt{\frac{\log\frac{1}{\delta}}{n}}\right)}\right). \tag{43}$$

Using equation 24, equation 26 and equation 43, we have

$$\left|\mathbb{E}_{\hat{P}_0^m}\left[\phi_\gamma^c\left(\boldsymbol{X}, y; \theta\right)\right] - \mathbb{E}_{P_0}\left[\phi_\gamma^c\left(\boldsymbol{X}, y; \theta\right)\right]\right| \leq \mathcal{O}\left(\gamma\sqrt{\frac{d}{m}\left(\alpha\lambda_{\max} + \sqrt{\frac{d}{n}} + \sqrt{\frac{\log\frac{1}{\delta}}{n}}\right)} + \sqrt{\frac{\log 1/\delta}{m}}\right), \tag{44}$$

for all $\theta \in \tilde{\Theta}$, where $\tilde{\Theta}$ is defined as in equation 25. Now, we need to demonstrate that the parameter $\theta^*$ that minimizes the expected robust loss also satisfies the constraint defined in equation 23. Given the data generating distribution in here, we know that $\theta^*$ also minimizes the expected non-robust loss. Therefore, we have

$$\theta^* = \underset{\theta\in\Theta}{\arg\min}\,\mathbb{E}_{P_0}\left[l\left(\boldsymbol{X}, y; \theta\right)\right], \tag{45}$$

where for the loss function, cost functions and distributions considered in here we have:

$$\theta^* = \underset{\theta\in\Theta}{\arg\min}\,\mathbb{P}_{u\sim\mathcal{N}\left(\langle\theta,\boldsymbol{\mu}_0\rangle, \sigma^2\|\theta\|_2^2\right)}\left(u \leq 0\right)$$

$$= \underset{\theta\in\Theta}{\arg\min}\,\mathcal{Q}\left(\frac{\langle\theta, \boldsymbol{\mu}_0\rangle^2}{\sigma^2\|\theta\|_2^2}\right)$$

$$= \frac{\boldsymbol{\mu}_0}{\|\boldsymbol{\mu}_0\|_2}. \tag{46}$$

In the setting of the Theorem, we also have:

$$\|\boldsymbol{\mu}_0 - \boldsymbol{\mu}_1\|_2 \leq \alpha.$$

Due to the Auxiliary Lemma 2, we know that if we set

$$\gamma' = \frac{1}{\hat{\lambda}_{\max}\log n + \mathcal{O}(\frac{d}{n})}, \quad s = 1 - \gamma'\left(\hat{\lambda}_{\max}\left(1 - \alpha\right) - 3\mathcal{O}\left(\sqrt{\frac{d}{n}}\right)\right), \tag{47}$$

then with high probability $\theta^* \in \tilde{\Theta}$ and the following holds:

$$\mathbb{E}_{(\boldsymbol{X}, y)\sim P_0}\left[\phi_\gamma^c\left(\boldsymbol{X}, y; \hat{\theta}^{\text{RSS}}\right)\right] \leq \mathbb{E}_{(\boldsymbol{X}, y)\sim \hat{P}_0^m}\left[\phi_\gamma^c\left(\boldsymbol{X}, y; \hat{\theta}^{\text{RSS}}\right)\right]$$

$$+\mathcal{O}\left(\gamma\sqrt{\frac{d}{m}\left(\alpha\lambda_{\max}+\sqrt{\frac{d}{n}}+\sqrt{\frac{\log\frac{1}{\delta}}{n}}\right)}+\sqrt{\frac{\log 1/\delta}{m}}\right)$$

$$\leq \mathbb{E}_{(\boldsymbol{X},y)\sim\hat{P}_0^m}\left[\phi_\gamma^c\left(\boldsymbol{X},y;\theta^*\right)\right]$$

$$+\mathcal{O}\left(\gamma\sqrt{\frac{d}{m}\left(\alpha\lambda_{\max}+\sqrt{\frac{d}{n}}+\sqrt{\frac{\log\frac{1}{\delta}}{n}}\right)}+\sqrt{\frac{\log 1/\delta}{m}}\right)$$

$$\leq \mathbb{E}_{(\boldsymbol{X},y)\sim P_0}\left[\phi_\gamma^c\left(\boldsymbol{X},y;\theta^*\right)\right]$$

$$+\mathcal{O}\left(\gamma\sqrt{\frac{d}{m}\left(\alpha\lambda_{\max}+\sqrt{\frac{d}{n}}+\sqrt{\frac{\log\frac{1}{\delta}}{n}}\right)}+\sqrt{\frac{\log 1/\delta}{m}}\right). \quad (48)$$

Now, let's consider a scenario where we split our labeled samples into two parts, each containing $m/2$ samples. We then ignore the labels of the second part and add these samples to the unlabeled set. In this situation, we have $m/2$ labeled samples and $n + m/2$ unlabeled samples and the bound above inequalities changes as follows:

$$\mathbb{E}_{(\boldsymbol{X},y)\sim P_0}\left[\phi_\gamma^c\left(\boldsymbol{X},y;\hat{\theta}^{\mathrm{RSS}}\right)\right] \leq \mathbb{E}_{(\boldsymbol{X},y)\sim P_0}\left[\phi_\gamma^c\left(\boldsymbol{X},y;\theta^*\right)\right]$$

$$+\mathcal{O}\left(\gamma\sqrt{\frac{2d}{m}\left(\alpha\lambda_{\max}+\sqrt{\frac{2d}{2n+m}}+\sqrt{\frac{2\log\frac{1}{\delta}}{2n+m}}\right)}+\sqrt{\frac{2\log 1/\delta}{m}}\right),$$

$$(49)$$

which completes the proof. $\square$

*Proof of Theorem 4.2.* Using Theorem 4.1 and Auxiliary Lemma 1, with probability more than $1 - \delta$ we have:

$$R\left(\hat{\theta}^{\mathrm{RSS}},P_0\right) \leq \mathbb{E}_{(\boldsymbol{X},y)\sim P_0}\left[\phi_\gamma^c\left(\boldsymbol{X},y;\hat{\theta}^{\mathrm{RSS}}\right)\right]$$

$$\leq \mathbb{E}_{(\boldsymbol{X},y)\sim P_0}\left[\phi_\gamma^c\left(\boldsymbol{X},y;\theta^*\right)\right]+\mathcal{O}\left(\gamma\sqrt{\frac{2d}{m}\left(\alpha\lambda_{\max}+\sqrt{\frac{2d+2\log\frac{1}{\delta}}{2n+m}}\right)}+\sqrt{\frac{2\log 1/\delta}{m}}\right)$$

$$\leq R\left(\theta^*,P_0\right)+\frac{e^{\frac{-\langle\theta^*,\mu_0\rangle^2}{2\sigma_0^2}}}{2\gamma\sigma_0\sqrt{2\pi}}+\mathcal{O}\left(\gamma\sqrt{\frac{2d}{m}\left(\alpha\lambda_{\max}+\sqrt{\frac{2d+2\log\frac{1}{\delta}}{2n+m}}\right)}+\sqrt{\frac{2\log 1/\delta}{m}}\right).$$

$$(50)$$

For the setting described in the statement of the theorem, we know that $\theta^* = \mu_0$ and $\hat{\lambda}_{\max} \leq \|\mu_1\|_2^2 + \sigma_1^2 + \sqrt{\frac{d}{n}}$. By setting

$$\gamma = \frac{e^{\frac{-\hat{\lambda}_{\max}}{4\sigma_0^2}}}{\sqrt{2\sigma_0\sqrt{2\pi}}}\left(\sqrt{\frac{2d}{m}\left(\alpha\hat{\lambda}_{\max}+\sqrt{\frac{2d+2\log\frac{1}{\delta}}{2n+m}}\right)}+\sqrt{2\frac{\log 1/\delta}{m}}\right)^{-1/4},$$

and subsequent substitution in equation 50, one has

$$R\left(\hat{\theta}^{\mathrm{RSS}},P_0\right) \leq R\left(\theta^*,P_0\right)$$

$$+ \mathcal{O}\left(\frac{e^{\frac{-\|\mu_0\|_2^2}{4\sigma_0^2}}}{\sqrt{2\sigma_0\sqrt{2\pi}}}\left(\left(\|\mu_1\|_2^2+\sigma_1^2\right)\frac{2d}{m}\alpha+\frac{2d}{m}\sqrt{\frac{2d+\log\frac{1}{\delta}}{2n+m}}\right)^{1/4}+\sqrt{\frac{2\log 1/\delta}{m}}\right). \quad (51)$$

$$\square$$

*Proof of Corollary 4.3.* Due to the result of Theorem 4.2, to have an advantage over the ERM method we should have:

$$\left( \frac{2d\alpha}{m} + \frac{2d}{m}\sqrt{\frac{2d}{2n+m}} \right)^{\frac{1}{4}} \leq \sqrt{\frac{d}{m}}. \tag{52}$$

If we have the following inequality it can be seen that the above inequality will be satisfied:

$$\alpha \leq \frac{d}{m}, \; n \geq \frac{m^2}{d}. \tag{53}$$

It can be seen that if we have $\alpha \leq \mathcal{O}\left(\frac{1}{d}\right)$ and $n > \Omega\left(d^3\right)$, then with high probability we have :

$$e_P\left(\hat{\theta}^{\mathrm{RSS}}\right) \leq \mathcal{O}\left(\frac{1}{m}\right)^{\frac{1}{4}} + \sqrt{\frac{\log\frac{1}{\delta}}{m}}. \tag{54}$$

Therefore if we aim to have an excess risk less than $\epsilon$ we should have at least $\max\left\{\frac{1}{\epsilon^4}, \frac{\log\frac{1}{\delta}}{\epsilon^2}\right\}$ samples which is independent from the dimension of data.

$\square$

*Proof of theorem 4.4.* From the manuscript we know that $\hat{\theta}^{\mathrm{RSS}}$ is the result of the following optimization problem:

$$\hat{\theta}^{\mathrm{RSS}} \triangleq \underset{\theta \in \Theta}{\arg\min} \left\{ \frac{1}{m}\sum_{i=1}^{m}\phi_\gamma\left(\boldsymbol{X}_i, y_i; \theta\right) + \frac{\lambda}{n}\sum_{j=1}^{n}\phi_{\gamma'}\left(\boldsymbol{X}_j', h_\theta\left(\boldsymbol{X}_j'\right); \theta\right) \right\}, \tag{55}$$

where we have $\Theta \triangleq \left\{\theta \in \mathbb{R}^d, \|\theta\|_2 = 1\right\}$. Like the proof of Theorem 4.1 if we consider the loss function mentioned in Theorem 4.1, along with the cost functions $c(\boldsymbol{X}, \boldsymbol{X}') = \|\boldsymbol{X} - \boldsymbol{X}'\|_2$ and $c'(\boldsymbol{X}, \boldsymbol{X}') = \|\boldsymbol{X} - \boldsymbol{X}'\|_2^2$, the Slater conditions will be satisfied. Hence, we have strong duality, and therefore, there exists $s \geq 0$ such that $\hat{\theta}^{\mathrm{RSS}}$ is also the result of the following optimization problem:

$$\hat{\theta}^{\mathrm{RSS}} = \underset{\theta: \frac{1}{n}\sum_{j=1}^{n}\phi_{\gamma'}\left(\boldsymbol{X}_j', h_\theta\left(\boldsymbol{X}_j'\right); \theta\right) \leq s}{\arg\min} \frac{1}{m}\sum_{i=1}^{m}\phi_\gamma\left(\boldsymbol{X}_i, y_i; \theta\right). \tag{56}$$

To prove the theorem, we need to establish an upper bound on the distance between the expected robust loss of $\hat{\theta}^{\mathrm{RSS}}$ and the best possible expected robust loss. To achieve this, we first provide a concentration bound for the maximum distance between the empirical and expected robust loss across all parameters with

$$\frac{1}{n}\sum_{j=1}^{n}\phi_{\gamma'}\left(\boldsymbol{X}_j', h_\theta\left(\boldsymbol{X}_j'\right); \theta\right) \leq s, \tag{57}$$

and then, we demonstrate that the parameter $\theta^\star$, which achieves the best possible robust loss, satisfies equation 57. Suppose that the loss function is bounded with $b$,

$$l\left(\boldsymbol{X}, y; \theta\right) \leq b, \; \forall \theta, \boldsymbol{X}, y,$$
$$\phi_\gamma^c\left(\boldsymbol{X}, y; \theta\right) \leq b, \; \forall c, \gamma, \theta, \boldsymbol{X}, y,$$

where the second inequality results from the first one. Therefore, $\frac{1}{m}\sum_{i=1}^{m}\phi_\gamma\left(\boldsymbol{X}_i, y_i; \theta\right)$ has the bounded difference property with parameter $b/m$, which results that

$$\Phi\left(P_0, S_0, \gamma, c\right) = \sup_{\theta \in \tilde{\Theta}} \left( \frac{1}{m}\sum_{i=1}^{m}\phi_\gamma\left(\boldsymbol{X}_i, y_i; \theta\right) - \mathbb{E}_{P_0}\left[\phi_\gamma^c\left(\boldsymbol{X}, y; \theta\right)\right] \right), \tag{58}$$

also has the bounded difference property with parameter $b/m$, where $\tilde{\Theta}$ is defined as

$$\tilde{\Theta} \triangleq \left\{ \theta : \frac{1}{n}\sum_{j=1}^{n}\phi_{\gamma'}\left(\boldsymbol{X}_j', h_\theta\left(\boldsymbol{X}_j'\right); \theta\right) \leq s, \|\theta\|_2 = 1 \right\}. \tag{59}$$

So, we have the following inequality with probability more than $1 - \delta$:

$$\Phi\left(P_0, S_0, \gamma, c\right) \leq \mathbb{E}\left[\Phi\left(P_0, S_0, \gamma, c\right)\right] + \sqrt{\frac{\log 1/\delta}{m}}, \quad \forall \theta \in \tilde{\Theta}. \tag{60}$$

In the above inequality we can give an upper-bound for the first term in the right hand side of the inequality as follows:

$$\mathbb{E}\left[\Phi\left(P_0, S_0, \gamma, c\right)\right] = \mathbb{E}_{S_0 \sim P_0^m}\left[\sup_{\theta \in \tilde{\Theta}}\left(\frac{1}{m}\sum_{i=1}^m \phi_\gamma\left(\boldsymbol{X}_i, y_i; \theta\right) - \mathbb{E}_{P_0}\left[\phi_\gamma^c\left(\theta; \boldsymbol{X}, y\right)\right]\right)\right]$$

$$\leq \mathbb{E}_{S_0, S_0'}\left[\sup_{\theta \in \tilde{\Theta}}\left(\frac{1}{m}\sum_{i=1}^m \phi_\gamma\left(\boldsymbol{X}_i, y_i; \theta\right) - \frac{1}{m}\sum_{i=1}^m \phi_\gamma\left(\boldsymbol{X}_i', y_i'; \theta\right)\right)\right]$$

$$\leq 2\mathbb{E}_{S_0, \epsilon}\left[\sup_{\theta \in \tilde{\Theta}}\frac{1}{m}\sum_{(\boldsymbol{X}_i, y_i) \in S_0}\epsilon_i \phi_\gamma^c\left(\theta; \boldsymbol{X}_i, y_i\right)\right]. \tag{61}$$

Consider the loss function mentioned in the Theorem 4.4, and the following cost functions $c, c'$

$$c(\boldsymbol{X}, \boldsymbol{X}') = \|\boldsymbol{X} - \boldsymbol{X}'\|_2, \quad c'(\boldsymbol{X}, \boldsymbol{X}') = \|\boldsymbol{X} - \boldsymbol{X}'\|_2^2.$$

With these loss and cost functions we can rewrite $\phi_\gamma^c$ and $\phi_{\gamma'}^{c'}$ as follows:

$$\phi_\gamma^c\left(\boldsymbol{X}, y; \theta\right) = \min\left\{1, \max\left\{0, 1 - \gamma y\langle\theta, \boldsymbol{X}\rangle\right\}\right\}, \tag{62}$$

$$\phi_{\gamma'}^{c'}\left(\boldsymbol{X}, h_\theta\left(\boldsymbol{X}\right); \theta\right) = \max\left\{0, 1 - \gamma'\langle\theta, \boldsymbol{X}\rangle^2\right\}. \tag{63}$$

As we can see $\phi_\gamma^c\left(\boldsymbol{X}, y; \theta\right)$ is a $\gamma$-lipschitz function of $y\langle\theta, \boldsymbol{X}\rangle$, therefore due to the Talagrand's lemma we can continue inequalities in equation 61 as follows:

$$\mathbb{E}\left[\Phi\left(P + 0, S_0, \gamma, c\right)\right] \leq \mathbb{E}_{S_0, \epsilon}\left[\sup_{\theta \in \tilde{\Theta}}\frac{1}{m}\sum_{(\boldsymbol{X}_i, y_i) \in S_0}\epsilon_i \phi_\gamma^c\left(\boldsymbol{X}_i, y_i; \theta\right)\right]$$

$$\leq \gamma\mathbb{E}_{S_0, \epsilon}\left[\sup_{\theta \in \tilde{\Theta}}\frac{1}{m}\sum_{(\boldsymbol{X}_i, y_i) \in S_0}\epsilon_i y_i\langle\theta, \boldsymbol{X}_i\rangle\right]$$

$$\leq \gamma\mathbb{E}_{S_0, \epsilon}\left[\sup_{\theta \in \tilde{\Theta}}\langle\theta, \frac{1}{m}\sum_{\boldsymbol{X}_i \in S_0}\epsilon_i y_i \boldsymbol{X}_i\rangle\right]. \tag{64}$$

Now suppose that we weaken the constraint in the above inequalities as follows:

$$s \geq \frac{1}{n}\sum_{j=1}^n \phi_{\gamma'}\left(\boldsymbol{X}_j', h_\theta\left(\boldsymbol{X}_j'\right); \theta\right)$$

$$\geq \mathbb{E}_{P_{1X, \theta}}\left[\phi_{\gamma'}^{c'}\left(\boldsymbol{X}, y; \theta\right)\right] - \frac{4\gamma'\left(\text{Tr}\left(\Sigma_1\right) + \|\boldsymbol{\mu}_1\|_2^2\right)}{\sqrt{n}} - \sqrt{\frac{\log\frac{1}{\delta}}{n}} \tag{65}$$

$$\geq \frac{4e^{-\frac{\langle\theta, \boldsymbol{\mu}_1\rangle^2}{2\theta^T\Sigma_1\theta}}}{3\gamma'} - \frac{4\gamma'\left(\text{Tr}\left(\Sigma_1\right) + \|\boldsymbol{\mu}_1\|_2^2\right)}{\sqrt{n}} - \sqrt{\frac{\log\frac{1}{\delta}}{n}}, \tag{66}$$

where we use Auxiliary Lemma 3 in the equation 65 and Auxiliary Lemma 4 in the equation 66. Suppose that we set $s$ as follows:

$$s = \inf_{\theta \in \Theta}\frac{1}{n}\sum_{\boldsymbol{X}_i \in S_1'}\phi_{\gamma'}^{c'}\left(\boldsymbol{X}_i, h_\theta\left(\boldsymbol{X}_i\right); \theta\right) + \frac{12\gamma'\text{Tr}\left(\hat{\Sigma}\left(S_1'\right)\right)}{\sqrt{n}} + 3\sqrt{\frac{\log\frac{1}{\delta}}{n}} + 16\sqrt{\frac{d}{n}} + \alpha, \tag{67}$$

where in the above equation $S_1'$ is generated in a same way but independent of $S_1$, and $\hat{\Sigma}\left(S_1'\right)$ is the sample covariance matrix of $S_1$. Therefore we have the following upper-bound for $s$:

$$s = \inf_{\theta \in \Theta}\frac{1}{n}\sum_{\boldsymbol{X}_i \in S_1'}\phi_{\gamma'}^{c'}\left(\boldsymbol{X}_i, h_\theta\left(\boldsymbol{X}_i\right); \theta\right) + \frac{12\gamma'\text{Tr}\left(\hat{\Sigma}\left(S_1'\right)\right)}{\sqrt{n}} + 3\sqrt{\frac{\log\frac{1}{\delta}}{n}} + 16\sqrt{\frac{d}{n}} + \alpha$$

$$\leq \inf_{\theta \in \Theta} \mathbb{E}_{P_{1X,\theta}} \left[ \phi_{\gamma'}^{c'} \left( \boldsymbol{X}, h_\theta \left( \boldsymbol{X}_i \right); \theta \right) \right] + \frac{24\gamma' \left( \operatorname{Tr} \left( \Sigma_1 \right) + \|\boldsymbol{\mu}_1\|_2^2 \right)}{\sqrt{n}} + 6\sqrt{\frac{\log \frac{1}{\delta}}{n}} + 32\sqrt{\frac{d}{n}} + \alpha \tag{68}$$

$$\leq \inf_{\theta \in \Theta} \frac{4e^{-\frac{\langle \theta, \boldsymbol{\mu}_1 \rangle^2}{2\theta^T \Sigma_1 \theta}}}{3\sqrt{\gamma'}} + \frac{24\gamma' \left( \operatorname{Tr} \left( \Sigma_1 \right) + \|\boldsymbol{\mu}_1\|_2^2 \right)}{\sqrt{n}} + 6\sqrt{\frac{\log \frac{1}{\delta}}{n}} + 32\sqrt{\frac{d}{n}} + \alpha$$

$$\leq \frac{4e^{-\frac{\boldsymbol{\mu}_1^T \Sigma_1^{-1} \boldsymbol{\mu}_1}{2}}}{3\sqrt{\gamma'}} + \frac{24\gamma' \left( \operatorname{Tr} \left( \Sigma_1 \right) + \|\boldsymbol{\mu}_1\|_2^2 \right)}{\sqrt{n}} + 6\sqrt{\frac{\log \frac{1}{\delta}}{n}} + 32\sqrt{\frac{d}{n}} + \alpha \tag{69}$$

where in the equation 68 we use the result of Auxiliary Lemma 4. In the equation 69 we insert $\theta = \Sigma_1^{-1} \boldsymbol{\mu}_1$ which minimizes $e^{-\frac{\langle \theta, \boldsymbol{\mu}_1 \rangle^2}{2\theta^T \Sigma_1 \theta}}$. We use equation 65 to equation 69, to define $\tilde{\Theta}_w$, which is a super-set for $\tilde{\Theta}$, as follows:

$$\tilde{\Theta}_w = \left\{ \theta \in \Theta : \frac{4e^{-\frac{\langle \theta, \boldsymbol{\mu}_1 \rangle^2}{2\theta^T \Sigma_1 \theta}}}{3\sqrt{\gamma'}} \leq \frac{4e^{-\frac{\boldsymbol{\mu}_1^T \Sigma_1^{-1} \boldsymbol{\mu}_1}{2}}}{3\sqrt{\gamma'}} + \frac{24\gamma' \left( \operatorname{Tr} \left( \Sigma_1 \right) + \|\boldsymbol{\mu}_1\|_2^2 \right)}{\sqrt{n}} + 6\sqrt{\frac{\log \frac{1}{\delta}}{n}} + 32\sqrt{\frac{d}{n}} + \alpha \right\}$$

$$= \left\{ \theta \in \Theta : \frac{\langle \theta, \boldsymbol{\mu}_1 \rangle^2}{\theta^T \Sigma_1 \theta} \geq \boldsymbol{\mu}_1 \Sigma_1^{-1} \boldsymbol{\mu}_1 - \right.$$
$$\left. \frac{e^{\frac{\boldsymbol{\mu}_1 \Sigma_1^{-1} \boldsymbol{\mu}_1}{2}} \left( 18\gamma'^2 \left( \operatorname{Tr} \left( \Sigma_1 \right) + \|\boldsymbol{\mu}_1\|_2^2 \right) + 5\gamma' \sqrt{\log \frac{1}{\delta}} + 24\sqrt{d} + 4\alpha\sqrt{n\gamma'} \right)}{2\sqrt{n}} \right\}$$

$$= \left\{ \theta \in \Theta : \boldsymbol{\mu}_1 \Sigma_1^{-1} \boldsymbol{\mu}_1 - \frac{\langle \theta, \boldsymbol{\mu}_1 \rangle^2}{\theta^T \Sigma_1 \theta} \leq \right.$$
$$\left. \frac{e^{\frac{\boldsymbol{\mu}_1 \Sigma_1^{-1} \boldsymbol{\mu}_1}{2}} \left( 18\gamma'^2 \left( \operatorname{Tr} \left( \Sigma_1 \right) + \|\boldsymbol{\mu}_1\|_2^2 \right) + 5\gamma' \sqrt{\log \frac{1}{\delta}} + 24\sqrt{d} + 4\alpha\sqrt{n\gamma'} \right)}{2\sqrt{n}} \right\}. \tag{70}$$

We can write $\theta = \frac{\Sigma_1^{-1} \boldsymbol{\mu}_1 + \epsilon \boldsymbol{v}}{\|\Sigma_1^{-1} \boldsymbol{\mu}_1 + \epsilon \boldsymbol{v}\|_2}$, where $v$ is a vector which its Euclidean norm is equal to 1 and belongs to the hyperplane orthogonal to $\Sigma_1^{-1} \boldsymbol{\mu}_1$, and $\epsilon \in \mathbb{R}$. If we use this new notation we can rewrite $\tilde{\Theta}_w$ as follows:

$$\tilde{\Theta}_w = \left\{ \theta = \frac{\Sigma_1^{-1} \boldsymbol{\mu}_1 + \epsilon \boldsymbol{v}}{\|\Sigma_1^{-1} \boldsymbol{\mu}_1 + \epsilon \boldsymbol{v}\|_2} : \epsilon^2 \boldsymbol{v}^T \left( \Sigma_1 - \frac{\boldsymbol{\mu}_1 \boldsymbol{\mu}_1^T}{\boldsymbol{\mu}_1^T \Sigma_1^{-1} \boldsymbol{\mu}_1} \right) \boldsymbol{v} \leq t, \ \boldsymbol{v}^T \Sigma_1^{-1} \boldsymbol{\mu}_1 = 0, \|\boldsymbol{v}\|_2 = 1 \right\}, \tag{71}$$

$$t = \frac{e^{\frac{\boldsymbol{\mu}_1 \Sigma_1^{-1} \boldsymbol{\mu}_1}{2}} \left( 18\gamma'^2 \left( \operatorname{Tr} \left( \Sigma_1 \right) + \|\boldsymbol{\mu}_1\|_2^2 \right) + 5\gamma' \sqrt{\log \frac{1}{\delta}} + 24\sqrt{d} + 4\alpha\sqrt{n\gamma'} \right)}{2\sqrt{n}} \tag{72}$$

where due to Auxiliary Lemma 7 we have:

$$\boldsymbol{v}^T \left( \Sigma_1 - \frac{\boldsymbol{\mu}_1 \boldsymbol{\mu}_1^T}{\boldsymbol{\mu}_1^T \Sigma_1^{-1} \boldsymbol{\mu}_1} \right) \boldsymbol{v} \geq \frac{\Delta \left( \Sigma_1 \right)}{d \kappa_1 \kappa_1'}, \tag{73}$$

where

$$\kappa_1 = \frac{\lambda_{\max} \left( \Sigma_1 \right)}{\lambda_{\min} \left( \Sigma_1 \right)}, \ \kappa_1' = \frac{\lambda_{\max} \left( \Sigma_1 \right)}{\Delta \left( \Sigma_1 \right)}$$
$$\Delta \left( \Sigma_1 \right) = \min \left\{ \lambda_i \left( \Sigma_1 \right) - \lambda_j \left( \Sigma_1 \right) \right\}, \ \forall i, j : \lambda_i \left( \Sigma_1 \right) \neq \lambda_j \left( \Sigma_1 \right).$$

Therefore we can weaken $\tilde{\Theta}_w$ as follows:

$$\tilde{\Theta}_w = \left\{ \theta = \frac{\Sigma_1^{-1} \boldsymbol{\mu}_1 + \epsilon \boldsymbol{v}}{\|\Sigma_1^{-1} \boldsymbol{\mu}_1 + \epsilon \boldsymbol{v}\|_2} : \epsilon \leq t', \ \boldsymbol{v}^T \Sigma_1^{-1} \boldsymbol{\mu}_1 = 0, \|\boldsymbol{v}\|_2 = 1 \right\}, \tag{74}$$

$$t' = \sqrt{\frac{d\kappa_1 \kappa_1'}{\Delta\left(\Sigma_1\right)} \frac{e^{\frac{\mu_1 \Sigma_1^{-1} \mu_1}{2}}\left(18\gamma'^2\left(\mathrm{Tr}\left(\Sigma_1\right) + \|\boldsymbol{\mu}_1\|_2^2\right) + 5\gamma'\sqrt{\log\frac{1}{\delta}} + 24\sqrt{d} + 4\alpha\sqrt{n}\gamma'\right)}{2\sqrt{n}}}. \quad (75)$$

Due to the definition of $\tilde{\Theta}_w$ in 74 and the fact that it is a super-set for $\tilde{\Theta}$ we can continue equation 64 as follows:

$$\mathbb{E}\left[\Phi\left(P + 0, S_0, \gamma, c\right)\right] \leq \gamma\mathbb{E}_{S_0,\epsilon}\left[\sup_{\theta\in\tilde{\Theta}}\langle\theta, \frac{1}{m}\sum_{\boldsymbol{X}_i\in S_0}\epsilon_i y_i \boldsymbol{X}_i\rangle\right]$$

$$\leq \gamma\mathbb{E}_{S_0,\epsilon}\left[\sup_{\theta\in\tilde{\Theta}_w}\langle\theta, \frac{1}{m}\sum_{\boldsymbol{X}_i\in S_0}\epsilon_i y_i \boldsymbol{X}_i\rangle\right]$$

$$\leq \gamma t'\sqrt{\frac{\lambda_{\max}\left(\Sigma_1\right)\mathbb{E}\left[\|\boldsymbol{X}\|_2^2\right]}{m\|\boldsymbol{\mu}_1\|_2}}$$

$$\leq \gamma t'\sqrt{\frac{\left(\|\boldsymbol{\mu}_1\|_2^2 + \mathrm{Tr}\left(\Sigma_1\right)\right)\lambda_{\max}\left(\Sigma_1\right)}{m\|\boldsymbol{\mu}_1\|_2}}, \quad (76)$$

Using Inequalities equation 76 and equation 60 we have:

$$\left|\frac{1}{m}\sum_{i=1}^m \phi_\gamma\left(\boldsymbol{X}_i, y_i; \theta\right) - \mathbb{E}_{P_0}\left[\phi_\gamma^c\left(\boldsymbol{X}, y\right); \theta\right]\right| \leq \gamma t'\sqrt{\frac{\left(\|\boldsymbol{\mu}_1\|_2^2 + \mathrm{Tr}\left(\Sigma_1\right)\right)\lambda_{\max}\left(\Sigma_1\right)}{m\|\boldsymbol{\mu}_1\|_2}} + \sqrt{\frac{\log\frac{1}{\delta}}{m}}. \quad (77)$$

Suppose that we define $\theta^*$ as follows:

$$\theta^* = \arg\min_{\theta\in\Theta}\mathbb{E}_{P_0}\left[l\left(\boldsymbol{X}, y; \theta\right)\right]. \quad (78)$$

For the loss function, cost functions and distributions considered in the Theorem 4.4 we have:

$$\theta^* = \arg\min_{\theta\in\Theta}\mathbb{P}_{u\sim\mathcal{N}\left(\langle\theta,\boldsymbol{\mu}_0\rangle, \theta^T\Sigma_0\theta\right)}\left(u \leq 0\right)$$

$$= \arg\min_{\theta\in\Theta}\mathcal{Q}\left(\frac{\langle\theta, \boldsymbol{\mu}_0\rangle^2}{\theta^T\Sigma_0\theta}\right)$$

$$= \frac{\Sigma_0^{-1}\boldsymbol{\mu}_0}{\|\Sigma_0^{-1}\boldsymbol{\mu}_0\|_2}. \quad (79)$$

In the setting of the theorem, we also have:

$$\|\boldsymbol{\mu}_0 - \boldsymbol{\mu}_1\|_2 \leq \alpha, \|\Sigma_1 - \Sigma_0\|_2 \leq \alpha$$

Due to the Lemma 6 we know that if we set $s$ and $\gamma'$ as

$$s = \inf_{\theta\in\Theta}\frac{1}{n}\sum_{\boldsymbol{X}_i\in S_1'}\phi_{\gamma'}^{c'}\left(\boldsymbol{X}_i, h_\theta\left(\boldsymbol{X}_i\right); \theta\right) + \frac{12\gamma'\mathrm{Tr}\left(\hat{\Sigma}\left(S_1'\right)\right)}{\sqrt{n}} + 2\sqrt{\frac{\log\frac{1}{\delta}}{n}} + 16\sqrt{\frac{d}{n}} + \alpha$$

$$\gamma' = 2e^{-\frac{\beta}{2}\sqrt{\hat{\lambda}_{\max}\left(\hat{\Sigma}\left(S_1'\right)\right)}} \quad (80)$$

then with high probability $\theta^* \in \tilde{\Theta}$, and we have the following:

$$\mathbb{E}_{P_0}\left[\phi_\gamma^c\left(\boldsymbol{X}, y; \hat{\theta}^{\mathrm{RSS}}\right)\right] \leq \mathbb{E}_{P_0}\left[\phi_\gamma^c\left(\boldsymbol{X}, y; \theta^*\right)\right] + \gamma t'\sqrt{\frac{\left(\|\boldsymbol{\mu}_1\|_2^2 + \mathrm{Tr}\left(\Sigma_1\right)\right)\lambda_{\max}\left(\Sigma_1\right)}{m\|\boldsymbol{\mu}_1\|_2}} + \sqrt{\frac{\log\frac{1}{\delta}}{m}}. \quad (81)$$

Using the result of Auxiliary Lemma 5 and the above inequality we have:

$$\mathbb{E}_{(\boldsymbol{X},y)\sim P_0}\left[l\left(\boldsymbol{X}, y; \hat{\theta}^{\mathrm{RSS}}\right)\right] \leq \mathbb{E}_{(\boldsymbol{X},y)\sim P_0}\left[l\left(\boldsymbol{X}, y; \theta^*\right)\right] + \frac{e^{-\frac{\langle\theta^*,\boldsymbol{\mu}_0\rangle^2}{2\theta^{*T}\Sigma_0\theta^*}}}{2\gamma}$$

$$+\gamma t'\sqrt{\frac{\left(\|\boldsymbol{\mu}_1\|_2^2 + \mathrm{Tr}\left(\Sigma_1\right)\right)\lambda_{\max}\left(\Sigma_1\right)}{m\|\boldsymbol{\mu}_1\|_2}} + \sqrt{\frac{\log\frac{1}{\delta}}{m}}. \tag{82}$$

Therefore if we set $\gamma$ as follows:

$$\gamma = \sqrt{\frac{\sqrt{m}}{2t'\lambda_{\max}\left(\hat{\Sigma}_1\left(S_1'\right)\right)}}, \tag{83}$$

then we have:

$$\mathbb{E}_{(\boldsymbol{X},y)\sim P_0}\left[l\left(\boldsymbol{X},y;\hat{\theta}^{\mathrm{RSS}}\right)\right] \leq \mathbb{E}_{(\boldsymbol{X},y)\sim P_0}\left[l\left(\boldsymbol{X},y;\theta^*\right)\right]$$

$$+ \sqrt{t'\frac{\lambda_{\max}\left(\Sigma_1\right)}{\sqrt{m}\|\boldsymbol{\mu}_1\|_2}} + \sqrt{\frac{\log\frac{1}{\delta}}{m}}. \tag{84}$$

Now, if we split our labeled samples into two equal parts, each containing $m/2$ samples, and combine the unlabeled set with the second part, we have $m/2$ labeled samples and $n+m/2$ unlabeled samples. When we set $t'$ from equation 75 in this new scenario, the proof will be complete. $\qquad\square$

## C  AUXILIARY LEMMAS

*Auxiliary Lemma* 1.  Consider the setting described in Theorem 4.1. let us define $f_{\Theta,P}\left(\theta,\gamma\right)$ as

$$f_{\Theta,P}\left(\theta,\gamma\right) = \mathbb{E}_{(\boldsymbol{X},y)\sim P}\left[\phi_\gamma^c\left(\boldsymbol{X},y;\theta\right)\right] - \mathbb{E}_{(\boldsymbol{X},y)\sim P}\left[\ell\left(\boldsymbol{X},y;\theta\right)\right]. \tag{85}$$

Then, for the loss and cost functions described in Theorem 4.1, $P = \mathcal{N}_{y\boldsymbol{\mu},\sigma}\left(\boldsymbol{X},y\right)$, and the class of linear classifiers with $|\theta|_2 = 1$, the function $f_{\Theta,P}\left(\theta,\gamma\right)$ satisfies the following inequality:

$$f_{\Theta,P}\left(\theta,\gamma\right) \leq \frac{e^{\frac{-\langle\theta,\mu\rangle^2}{2\sigma^2}}}{2\gamma\sigma\sqrt{2\pi}}. \tag{86}$$

*Proof.*  For the setting described in the lemma we have:

$$u = y\langle\theta,\boldsymbol{X}\rangle \sim \mathcal{N}\left(\langle\theta,\mu\rangle,\sigma^2\right). \tag{87}$$

Therefore we can calculate an upper-bound for $f_{\Theta,P}\left(\theta,\gamma\right)$ as follows:

$$\mathbb{E}_{(\boldsymbol{X},y)\sim P}\left[\phi_\gamma^c\left(\boldsymbol{X},y;\theta\right)\right] - \mathbb{E}_{(\boldsymbol{X},y)\sim P}\left[l\left(\boldsymbol{X},y;\theta\right)\right] = \int_{u=0}^{1/\gamma}\frac{1}{\sqrt{2\pi\sigma^2}}\left(1-\gamma u\right)e^{-\frac{(u-\langle\theta,\mu\rangle)^2}{2\sigma^2}}\,du$$

$$\leq \int_{u=0}^{1/\gamma}\frac{1}{\sqrt{2\pi\sigma^2}}\left(1-\gamma u\right)e^{-\frac{\langle\theta,\mu\rangle^2}{2\sigma^2}}\,du$$

$$\leq \frac{e^{-\frac{\langle\theta,\mu\rangle^2}{2\sigma^2}}}{2\gamma\sigma\sqrt{2\pi}}. \tag{88}$$

And the proof is complete. $\qquad\square$

*Auxiliary Lemma* 2.  Consider the setting in Theorem 4.1 if we define $\theta^*$ as follows:

$$\theta^* \triangleq \frac{\boldsymbol{\mu}_0}{\|\boldsymbol{\mu}_0\|_2}, \tag{89}$$

then with high probability we have:

$$\frac{1}{n}\sum_{j=1}^n \phi_{\gamma'}\left(\boldsymbol{X}_j', h_\theta\left(\boldsymbol{X}_j'\right);\theta\right) \leq s,$$

As long as we set:

$$\gamma' = \frac{1}{\hat{\lambda}_{\max}\log n + \mathcal{O}(\frac{d}{n})}, \quad s = 1 - \gamma'\left(\hat{\lambda}_{\max}\left(1-\alpha\right) - 3\mathcal{O}\left(\sqrt{\frac{d}{n}}\right)\right) \tag{90}$$

*Proof.* Due to the boundedness of $\phi_{\gamma'}^{c'}$ with probability more than $1 - \delta$ we have the following inequality:

$$\frac{1}{n} \sum_{\boldsymbol{X}_i' \in S_1} \phi_{\gamma'}^{c'} \left( \boldsymbol{X}_i', h_{\theta^*} \left( \boldsymbol{X}_i' \right); \theta^* \right) \leq \mathbb{E}_{\boldsymbol{X} \sim P_{1X}} \left[ \phi_{\gamma'}^{c'} \left( \boldsymbol{X}, h_{\theta^*} \left( \boldsymbol{X} \right); \theta^* \right) \right] + \sqrt{\frac{\log \frac{1}{\delta}}{n}} \tag{91}$$

$$= \mathbb{E}_{\boldsymbol{X} \sim P_{1X}} \left[ \max \left\{ 0, 1 - \gamma' \langle \theta^*, \boldsymbol{X} \rangle^2 \right\} \right] + \sqrt{\frac{\log \frac{1}{\delta}}{n}}, \tag{92}$$

where we have the following upper-bound for the first term in 92:

$$\mathbb{E}_{\boldsymbol{X} \sim P_{1X}} \left[ \max \left\{ 0, 1 - \gamma' \langle \theta^*, \boldsymbol{X} \rangle^2 \right\} \right] = \mathbb{E}_{\boldsymbol{X} \sim P_{1X}} \left[ 1 - \gamma' \langle \theta^*, \boldsymbol{X} \rangle^2 \right]$$

$$- \int_{\langle \theta^*, \boldsymbol{X} \rangle^2 \geq \frac{1}{\gamma'}} 1 - \gamma' \langle \theta^*, \boldsymbol{X} \rangle^2 \, dP_{1X}$$

$$= \mathbb{E}_{\boldsymbol{X} \sim P_{1X}} \left[ 1 - \gamma' \langle \theta^*, \boldsymbol{X} \rangle^2 \right]$$

$$- \frac{1}{2} \int_{\langle \theta^*, \boldsymbol{X} \rangle^2 \geq \frac{1}{\gamma'}} 1 - \gamma' \langle \theta^*, \boldsymbol{X} \rangle^2 \, \mathcal{N} \left( \boldsymbol{\mu_1}, \sigma_1^2 I \right)$$

$$- \frac{1}{2} \int_{\langle \theta^*, \boldsymbol{X} \rangle^2 \geq \frac{1}{\gamma'}} 1 - \gamma' \langle \theta^*, \boldsymbol{X} \rangle^2 \, \mathcal{N} \left( -\boldsymbol{\mu_1}, \sigma_1^2 I \right)$$

$$\leq \mathbb{E}_{\boldsymbol{X} \sim P_{1X}} \left[ 1 - \gamma' \langle \theta^*, \boldsymbol{X} \rangle^2 \right]$$

$$- \int_{u \geq \frac{1}{\sigma_1 \sqrt{\gamma'}} - \frac{\langle \theta^*, \boldsymbol{\mu_1} \rangle}{\sigma_1}} 1 - \gamma' \left( \langle \theta^*, \boldsymbol{\mu_1} \rangle + \sigma_1 u \right)^2 \, \mathcal{N} \left( 0, \sigma_1^2 \right)$$

$$- \int_{u \leq -\frac{1}{\sigma_1 \sqrt{\gamma'}} + \frac{\langle \theta^*, \boldsymbol{\mu_1} \rangle}{\sigma_1}} 1 - \gamma' \left( -\langle \theta^*, \boldsymbol{\mu_1} \rangle + \sigma_1 u \right)^2 \, \mathcal{N} \left( 0, \sigma_1^2 \right). \tag{93}$$

Now if we set

$$\gamma' \leq \frac{1}{\left( \langle \theta^*, \boldsymbol{\mu_1} \rangle + \sigma_1 \sqrt{\log n} \right)^2} \tag{94}$$

we can continue inequalities in 93 as follows:

$$\mathbb{E}_{\boldsymbol{X} \sim P_{1X}} \left[ \max \left\{ 0, 1 - \gamma' \langle \theta^*, \boldsymbol{X} \rangle^2 \right\} \right] \leq \mathbb{E}_{\boldsymbol{X} \sim P_{1X}} \left[ 1 - \gamma' \langle \theta^*, \boldsymbol{X} \rangle^2 \right] + \frac{1}{\sqrt{n}}$$

$$= 1 - \gamma' \left( \langle \theta^*, \boldsymbol{\mu_1} \rangle^2 + \sigma_1^2 \right) + \frac{1}{\sqrt{n}}. \tag{95}$$

However, since we lack access to $\mu_1$, $\sigma_1$, or $\theta^*$, we cannot directly obtain $\gamma'$ from equation 94. Nevertheless, a suitable choice for $\gamma'$ that satisfies equation 94 and can be determined from samples is:

$$\gamma' = \frac{1}{\hat{\lambda}_{\max} \log n + \mathcal{O}(\frac{d}{n})}, \tag{96}$$

where $\hat{\lambda}_{\max}$ is the maximum eigenvalue of the sample covariance matrix, $\hat{\Sigma}$, of a set of $n$ unlabeled samples. Now from Equations 91 and 95 we have:

$$\frac{1}{n} \sum_{j=1}^{n} \phi_{\gamma'} \left( \boldsymbol{X}_j', h_\theta \left( \boldsymbol{X}_j' \right); \theta \right) \leq 1 - \gamma' \left( \langle \theta^*, \boldsymbol{\mu_1} \rangle^2 + \sigma_1^2 \right) + 2 \sqrt{\frac{\log \frac{1}{\delta}}{n}}. \tag{97}$$

To complete the proof we should show that :

$$1 - \gamma' \left( \langle \theta^*, \boldsymbol{\mu_1} \rangle^2 + \sigma_1^2 \right) + 2 \sqrt{\frac{\log \frac{1}{\delta}}{n}} \leq s, \tag{98}$$

where due to the setting of this lemma we only need to set:

$$s = 1 - \gamma' \left( \hat{\lambda}_{\max} (1 - \alpha) - 3\mathcal{O} \left( \sqrt{\frac{d}{n}} \right) \right). \tag{99}$$

$\square$

*Auxiliary Lemma* 3. Consider the setting described in Theorem 4.4, if we define the function $\phi_{\gamma'}^{c'}$ as follows

$$\phi_{\gamma'}^{c'} (\boldsymbol{X}, h_\theta (\boldsymbol{X}) ; \theta) = \max \left\{ 0, 1 - \gamma' \langle \theta, \boldsymbol{X} \rangle^2 \right\}, \tag{100}$$

then with probability greater than $1 - \delta$, for $\forall \theta \in \Theta$, with $\|\theta\|_2 = 1$ we have:

$$\left| \frac{1}{n} \sum_{i=1}^{n} \phi_{\gamma'}^{c'} (\boldsymbol{X}_i, h_\theta (\boldsymbol{X}_i) ; \theta) - \mathbb{E} \left[ \phi_{\gamma'}^{c'} (\boldsymbol{X}, h_\theta (\boldsymbol{X}) ; \theta) \right] \right| \leq \frac{4\gamma' \left( \operatorname{Tr} (\Sigma_1) + \|\boldsymbol{\mu}_1\|_2^2 \right)}{\sqrt{n}} + \sqrt{\frac{\log \frac{1}{\delta}}{n}}. \tag{101}$$

*Proof.* From the definition of $\phi_{\gamma'}^{c'}$ we know that this function is bounded between 0 and 1, therefore the following function has the bounded difference property with parameter $\frac{1}{n}$

$$\Phi = \sup_{\theta \in \Theta} \left| \frac{1}{n} \sum_{i=1}^{n} \phi_{\gamma'}^{c'} (\boldsymbol{X}_i, h_\theta (\boldsymbol{X}_i) ; \theta) - \mathbb{E} \left[ \phi_{\gamma'}^{c'} (\boldsymbol{X}, h_\theta (\boldsymbol{X}) ; \theta) \right] \right|. \tag{102}$$

So we can write the McDiarmid's inequality for this function as follows:

$$\Phi \leq \mathbb{E} [\Phi] + \sqrt{\frac{\log \frac{1}{\delta}}{n}}. \tag{103}$$

Now we try to give an upper-bound for the first term in the right-hand side of the above inequality:

$$
\begin{aligned}
\mathbb{E} [\Phi] =& \mathbb{E}_{\{\boldsymbol{X}_i\}_1^n} \left[ \sup_{\theta \in \Theta} \left| \frac{1}{n} \sum_{i=1}^{n} \phi_{\gamma'}^{c'} (\boldsymbol{X}_i, h_\theta (\boldsymbol{X}_i) ; \theta) - \mathbb{E} \left[ \phi_{\gamma'}^{c'} (\boldsymbol{X}, h_\theta (\boldsymbol{X}) ; \theta) \right] \right| \right] \\
\leq& \mathbb{E}_{\{\boldsymbol{X}_i\}_1^n, \{\boldsymbol{X}_i'\}_1^n} \left[ \sup_{\theta \in \Theta} \left| \frac{1}{n} \sum_{i=1}^{n} \phi_{\gamma'}^{c'} (\boldsymbol{X}_i, h_\theta (\boldsymbol{X}_i) ; \theta) - \frac{1}{n} \sum_{i=1}^{n} \phi_{\gamma'}^{c'} (\boldsymbol{X}_i', h_\theta (\boldsymbol{X}_i') ; \theta) \right| \right] \\
\leq& \mathbb{E}_{\{\boldsymbol{X}_i\}_1^n, \{\boldsymbol{X}_i'\}_1^n, \{\epsilon_i\}_1^n} \left[ \sup_{\theta \in \Theta} \epsilon_i \left( \frac{1}{n} \sum_{i=1}^{n} \phi_{\gamma'}^{c'} (\boldsymbol{X}_i, h_\theta (\boldsymbol{X}_i) ; \theta) - \frac{1}{n} \sum_{i=1}^{n} \phi_{\gamma'}^{c'} (\boldsymbol{X}_i', h_\theta (\boldsymbol{X}_i') ; \theta) \right) \right]
\end{aligned}
\tag{104}
$$

$$
\begin{aligned}
\leq& 2\mathbb{E}_{\{\boldsymbol{X}_i\}_1^n, \{\epsilon_i\}_1^n} \left[ \sup_{\theta \in \Theta} \frac{1}{n} \sum_{i=1}^{n} \epsilon_i \phi_{\gamma'}^{c'} (\boldsymbol{X}_i, h_\theta (\boldsymbol{X}_i) ; \theta) \right] \\
\leq& 2\gamma' \mathbb{E}_{\{\boldsymbol{X}_i\}_1^n, \{\epsilon_i\}_1^n} \left[ \sup_{\theta \in \Theta} \frac{1}{n} \sum_{i=1}^{n} \epsilon_i \langle \theta, \boldsymbol{X}_i \rangle^2 \right] \\
=& 2\gamma' \mathbb{E}_{\{\boldsymbol{X}_i\}_1^n, \{\epsilon_i\}_1^n} \left[ \sup_{\theta \in \Theta} \theta^T \left( \frac{1}{n} \sum_{i=1}^{n} \epsilon_i \boldsymbol{X}_i \boldsymbol{X}_i^T \right) \theta \right] \\
\leq& 2\gamma' \mathbb{E}_{\{\boldsymbol{X}_i\}_1^n, \{\epsilon_i\}_1^n} \left[ \lambda_{\max} \left( \frac{1}{n} \sum_{i=1}^{n} \epsilon_i \boldsymbol{X}_i \boldsymbol{X}_i^T \right) \right] \\
\leq& 2\gamma' \sqrt{\mathbb{E}_{\{\boldsymbol{X}_i\}_1^n, \{\epsilon_i\}_1^n} \left[ \operatorname{Tr} \left( \frac{1}{n} \sum_{i=1}^{n} \epsilon_i \boldsymbol{X}_i \boldsymbol{X}_i^T \right)^T \left( \frac{1}{n} \sum_{i=1}^{n} \epsilon_i \boldsymbol{X}_i \boldsymbol{X}_i^T \right) \right]} \\
\leq& 2\gamma' \sqrt{\mathbb{E}_{\{\boldsymbol{X}_i\}_1^n} \left[ \frac{1}{n^2} \sum_{i=1}^{n} \|\boldsymbol{X}_i\|_2^4 \right]}
\end{aligned}
$$

$$\leq \frac{2\gamma'}{\sqrt{n}} \sqrt{\mathbb{E}_{\boldsymbol{X}} \left[ \|\boldsymbol{X}\|_2^4 \right]}$$

$$\leq \frac{4\gamma' \left( \text{Tr} \left( \Sigma_1 \right) + \|\boldsymbol{\mu}_1\|_2^2 \right)}{\sqrt{n}}, \tag{105}$$

where $\epsilon_i$s in equation 104 are Rademacher's random variables, and the last inequality in equation 105 is results from the fact that $\boldsymbol{X}_i$s are sampled from distribution $P_1$ defined in Theorem 4.4 of the manuscript. And the prood is complete. $\square$

*Auxiliary Lemma* 4. Consider the setting described in Theorem 4.4, if we define the function $\phi_{\gamma'}^{c'}$ as follows

$$\phi_{\gamma'}^{c'} \left( \boldsymbol{X}, h_\theta \left( \boldsymbol{X} \right); \theta \right) = \max \left\{ 0, 1 - \gamma' \langle \theta, \boldsymbol{X} \rangle^2 \right\}, \tag{106}$$

then we have the following upper-bound and lower-bound for the expected value of $\phi_{\gamma'}^{c'}$ when $\boldsymbol{X}$ sampled from the distribution $P_1$ defined in Theorem 4.4 of the manuscript:

$$\frac{4e^{-\frac{\langle\theta,\boldsymbol{\mu}_1\rangle^2}{2\theta^T\Sigma_1\theta}}}{3\sqrt{\gamma'}} \left( 1 - \frac{1}{\sqrt{\gamma'\theta^T\Sigma_1\theta}} \right) \leq \mathbb{E}\left[ \phi_{\gamma'}^{c'} \left( \boldsymbol{X}, h_\theta \left( \boldsymbol{X} \right); \theta \right) \right] \leq \frac{4e^{-\frac{\langle\theta,\boldsymbol{\mu}_1\rangle^2}{2\theta^T\Sigma_1\theta}}}{3\sqrt{\gamma'}} \left( 1 + \frac{1}{\sqrt{\gamma'\theta^T\Sigma_1\theta}} \right). \tag{107}$$

*Proof.* For the expected value of $\phi_{\gamma'}^{c'}$ we can write:

$$\mathbb{E}\left[ \phi_{\gamma'}^{c'} \left( \boldsymbol{X}, h_\theta \left( \boldsymbol{X} \right); \theta \right) \right] = \max \left\{ 0, 1 - \gamma' \langle \theta, \boldsymbol{X} \rangle^2 \right\}$$

$$= \int_{-\frac{1}{\sqrt{\gamma'}} \leq \langle\theta,\boldsymbol{X}\rangle \leq \frac{1}{\sqrt{\gamma'}}} 1 - \gamma' \langle \theta, \boldsymbol{X} \rangle^2 \mathcal{N} \left( \boldsymbol{\mu}_1, \Sigma_1 \right)$$

$$= \int_{-\frac{1}{\sqrt{\gamma'\theta^T\Sigma_1\theta}} - \frac{\langle\theta,\boldsymbol{\mu}_1\rangle}{\sqrt{\theta^T\Sigma_1\theta}}}^{\frac{1}{\sqrt{\gamma'\theta^T\Sigma_1\theta}} - \frac{\langle\theta,\boldsymbol{\mu}_1\rangle}{\sqrt{\theta^T\Sigma_1\theta}}} 1 - \gamma' \left( \langle \theta, \boldsymbol{\mu}_1 \rangle + u\sqrt{\theta^T\Sigma_1\theta} \right)^2 \mathcal{N} \left( 0, 1 \right). \tag{108}$$

Where we have the following upper-bound for the expected value if $\gamma' >> \frac{1}{\langle\theta,\boldsymbol{\mu}_1\rangle^2}$ :

$$\mathbb{E}\left[ \phi_{\gamma'}^{c'} \left( \boldsymbol{X}, h_\theta \left( \boldsymbol{X} \right); \theta \right) \right] \leq \frac{4}{3\sqrt{\gamma'}} \mathcal{Q} \left( \frac{\langle \theta, \boldsymbol{\mu}_1 \rangle}{\sqrt{\theta^T\Sigma_1\theta}} - \frac{1}{\sqrt{\gamma'\theta^T\Sigma_1\theta}} \right)$$

$$\leq \frac{4e^{-\frac{\langle\theta,\boldsymbol{\mu}_1\rangle^2}{2\theta^T\Sigma_1\theta}}}{3\sqrt{\gamma'}} \left( 1 + \frac{1}{\sqrt{\gamma'\langle\theta,\boldsymbol{\mu}_1\rangle^2}} \right). \tag{109}$$

Now without loss of generality we assume that the largest eigenvalue of $\Sigma_1$ is less than or equal to 1, then we have the following lower-bound if $\gamma' >> \frac{1}{\langle\theta,\boldsymbol{\mu}_1\rangle^2}$:

$$\mathbb{E}\left[ \phi_{\gamma'}^{c'} \left( \boldsymbol{X}, h_\theta \left( \boldsymbol{X} \right); \theta \right) \right] \geq \frac{4e^{-\frac{\langle\theta,\boldsymbol{\mu}_1\rangle^2}{2\theta^T\Sigma_1\theta}}}{3\sqrt{\gamma'\theta^T\Sigma_1\theta}} \left( 1 - \frac{1}{\sqrt{\gamma'\langle\theta,\boldsymbol{\mu}_1\rangle^2}} \right)$$

$$\geq \frac{4e^{-\frac{\langle\theta,\boldsymbol{\mu}_1\rangle^2}{2\theta^T\Sigma_1\theta}}}{3\sqrt{\gamma'}} \left( 1 - \frac{1}{\sqrt{\gamma'\langle\theta,\boldsymbol{\mu}_1\rangle^2}} \right). \tag{110}$$

$\square$

*Auxiliary Lemma* 5. Consider any distribution $P$, loss function $\ell$, transportation cost $c$, and $\theta \in \Theta$. let us define $f_{\Theta,P}(\theta,\gamma)$ as

$$f_{\Theta,P}(\theta,\gamma) = \mathbb{E}_{(\boldsymbol{X},y)\sim P} \left[ \phi_\gamma^c \left( \boldsymbol{X}, y; \theta \right) \right] - \mathbb{E}_{(\boldsymbol{X},y)\sim P} \left[ \ell \left( \boldsymbol{X}, y; \theta \right) \right]. \tag{111}$$

Then, for the loss and cost functions described in Section 4, $P = P_0$, where $P_0$ described in Theorem 4.4, in the manuscript, and the class of linear classifiers with $|\theta|_2 = 1$, the function $f_{\Theta,P}(\theta,\gamma)$ satisfies the following inequality:

$$f_{\Theta,P}(\theta,\gamma) \leq \frac{e^{-\frac{\langle\theta,\boldsymbol{\mu}_0\rangle^2}{2\theta^T\Sigma_0\theta}}}{2\gamma}. \tag{112}$$

*Proof.* For the setting described in the lemma we have:

$$u = y\langle \theta, \boldsymbol{X} \rangle \sim \mathcal{N}\left( \langle \theta, \boldsymbol{\mu}_0 \rangle, \theta^T \Sigma_0 \theta \right). \tag{113}$$

Therefore we can calculate an upper-bound for $f_{\Theta,P}(\theta, \gamma)$ as follows:

$$\mathbb{E}_{(\boldsymbol{X},y)\sim P_0}\left[ \phi_\gamma^c\left( \boldsymbol{X}, y; \theta \right) \right] - \mathbb{E}_{(\boldsymbol{X},y)\sim P_0}\left[ l\left( \boldsymbol{X}, y; \theta \right) \right] = \int_{u=0}^{1/\gamma} \frac{1}{\sqrt{2\pi\theta^T\Sigma_0\theta}}\left( 1 - \gamma u \right) e^{\frac{(u-\langle \theta, \boldsymbol{\mu}_0 \rangle)^2}{2\theta^T\Sigma_0\theta}} \, du$$

$$\leq \frac{e^{-\frac{\langle \theta, \boldsymbol{\mu}_0 \rangle^2}{2\theta^T\Sigma_0\theta}}}{2\gamma}. \tag{114}$$

And the proof is complete. □

*Auxiliary Lemma* 6. Consider the setting in Theorem 4.4 if we define $\theta^*$ as follows:

$$\theta^* \triangleq \frac{\Sigma_0^{-1}\boldsymbol{\mu}_0}{\|\Sigma_0^{-1}\boldsymbol{\mu}_0\|_2}, \tag{115}$$

then with high probability we have:

$$\frac{1}{n}\sum_{i=1}^n \phi_{\gamma'}^{c'}\left( \boldsymbol{X}_i, h_\theta\left( \boldsymbol{X}_i \right); \theta \right) \leq s,$$

as long as we set:

$$s = \inf_{\theta \in \Theta} \frac{1}{n}\sum_{\boldsymbol{X}_i \in S_1'} \phi_{\gamma'}^{c'}\left( \boldsymbol{X}_i, h_\theta\left( \boldsymbol{X}_i \right); \theta \right) + \frac{12\gamma' \operatorname{Tr}\left( \hat{\Sigma}\left( S_1' \right) \right)}{\sqrt{n}} + 2\sqrt{\frac{\log \frac{1}{\delta}}{n}} + 16\sqrt{\frac{d}{n}} + \alpha$$

$$\gamma' = 2e^{-\frac{\beta}{2}\sqrt{\hat{\lambda}_{\max}\left( \hat{\Sigma}\left( S_1' \right) \right)}} \tag{116}$$

*Proof.* To prove the lemma we first give a lower-bound for $s$.

$$s = \inf_{\theta \in \Theta} \frac{1}{n}\sum_{\boldsymbol{X}_i \in S_1'} \phi_{\gamma'}^{c'}\left( \boldsymbol{X}_i, h_\theta\left( \boldsymbol{X}_i \right); \theta \right) + \frac{12\gamma' \operatorname{Tr}\left( \hat{\Sigma}\left( S_1' \right) \right)}{\sqrt{n}} + 3\sqrt{\frac{\log \frac{1}{\delta}}{n}} + 16\sqrt{\frac{d}{n}} + \alpha$$

$$\geq \inf_{\theta \in \Theta} \mathbb{E}_{P_{1X,\theta}}\left[ \phi_{\gamma'}^{c'}\left( \boldsymbol{X}, y; \theta \right) \right] + \frac{4\gamma'\left( \operatorname{Tr}\left( \Sigma_1 \right) + \|\boldsymbol{\mu}_1\|_2^2 \right)}{\sqrt{n}} + \sqrt{\frac{\log \frac{1}{\delta}}{n}} + \alpha \tag{117}$$

$$\geq \frac{4e^{-\frac{\boldsymbol{\mu}_1^T\Sigma_1^{-1}\boldsymbol{\mu}_1}{2}}}{3\sqrt{\gamma'}} + \frac{4\gamma'\left( \operatorname{Tr}\left( \Sigma_1 \right) + \|\boldsymbol{\mu}_1\|_2^2 \right)}{\sqrt{n}} + \sqrt{\frac{\log \frac{1}{\delta}}{n}} + \alpha, \tag{118}$$

where inequality 117 is due to Auxiliary Lemma 3. To complete the proof we give an upper-bound for $\frac{1}{n}\sum_{i=1}^n \phi_{\gamma'}^{c'}\left( \boldsymbol{X}_i, h_\theta\left( \boldsymbol{X}_i \right); \theta^* \right)$:

$$\frac{1}{n}\sum_{i=1}^n \phi_{\gamma'}^{c'}\left( \boldsymbol{X}_i, h_\theta\left( \boldsymbol{X}_i \right); \theta^* \right) \leq \mathbb{E}_{P_{1X,\theta}}\left[ \phi_{\gamma'}^{c'}\left( \boldsymbol{X}, y; \theta^* \right) \right] + \frac{4\gamma'\left( \operatorname{Tr}\left( \Sigma_1 \right) + \|\boldsymbol{\mu}_1\|_2^2 \right)}{\sqrt{n}} + \sqrt{\frac{\log \frac{1}{\delta}}{n}}$$

$$\leq \frac{4e^{-\frac{\langle \theta^*, \boldsymbol{\mu}_1 \rangle^2}{2\theta^{*T}\Sigma_1\theta^*}}}{3\sqrt{\gamma'}} + \frac{4\gamma'\left( \operatorname{Tr}\left( \Sigma_1 \right) + \|\boldsymbol{\mu}_1\|_2^2 \right)}{\sqrt{n}} + \sqrt{\frac{\log \frac{1}{\delta}}{n}}. \tag{119}$$

From equation 119 and equation 118, we can complete the proof by setting $\gamma'$ as follows:

$$\gamma' \geq \frac{16}{9}\left( \frac{e^{-\frac{\langle \theta^*, \boldsymbol{\mu}_1 \rangle^2}{2\theta^{*T}\Sigma_1\theta^*}} - e^{-\frac{\boldsymbol{\mu}_1^T\Sigma_1^{-1}\boldsymbol{\mu}_1}{2}}}{\alpha} \right)^2 \approx 2e^{-\boldsymbol{\mu}_1^T\Sigma_1^{-1}\boldsymbol{\mu}_1}, \tag{120}$$

where in the above inequalities we suppose that $\alpha$ is small enough. Suppose that we know $\|\boldsymbol{\mu}_1\|_2 \geq \beta\lambda_{\max}\left( \Sigma_1 \right)$, then a good choice for $\gamma'$ will be:

$$\gamma' = 2e^{-\frac{\beta}{2}\sqrt{\hat{\lambda}_{\max}\left( \hat{\Sigma}\left( S_1' \right) \right)}} \tag{121}$$

□

*Auxiliary Lemma* 7. Consider a positive definite $d \times d$ matrix, $\Sigma$, and a vector $\boldsymbol{\mu} \in \mathbb{R}^d$. For and vector $\boldsymbol{v} \in \mathbb{R}^d$, that we know:

$$\|\boldsymbol{v}\|_2 = 1, \ \boldsymbol{v}^T \Sigma_1^{-1} \boldsymbol{\mu} = 0, \tag{122}$$

we have the following:

$$\boldsymbol{v}^T \left( \Sigma - \frac{\boldsymbol{\mu}\boldsymbol{\mu}^T}{\boldsymbol{\mu}^T \Sigma^{-1} \boldsymbol{\mu}} \right) \boldsymbol{v} \geq \frac{\lambda_{\min}\Delta^2}{d\lambda_{\max}^2}. \tag{123}$$

*Proof.* Suppose that we write $\boldsymbol{v}$ as follows:

$$\boldsymbol{v} = \psi_1 \hat{\boldsymbol{\mu}} + \psi_1 \hat{\boldsymbol{\mu}}^\perp, \ \psi_1^2 + \psi_2^2 = 1, \tag{124}$$

where $\hat{\boldsymbol{\mu}}$ is a unit norm vector in the direction of $\boldsymbol{\mu}$, and $\hat{\boldsymbol{\mu}}^\perp$ is a unit norm vector perpendicular to $\boldsymbol{\mu}$. If we use this new description of $\boldsymbol{v}$ we have:

$$\boldsymbol{v}^T \left( \Sigma - \frac{\boldsymbol{\mu}\boldsymbol{\mu}^T}{\boldsymbol{\mu}^T \Sigma^{-1} \boldsymbol{\mu}} \right) \boldsymbol{v} = \psi_1^2 \hat{\boldsymbol{\mu}}^T \left( \Sigma - \frac{\boldsymbol{\mu}\boldsymbol{\mu}^T}{\boldsymbol{\mu}^T \Sigma^{-1} \boldsymbol{\mu}} \right) \hat{\boldsymbol{\mu}} + \left( \psi_2^2 + 2\psi_1\psi_2 \right) \lambda_{\min}(\Sigma)$$

$$\geq \min \left\{ \hat{\boldsymbol{\mu}}^T \left( \Sigma - \frac{\boldsymbol{\mu}\boldsymbol{\mu}^T}{\boldsymbol{\mu}^T \Sigma^{-1} \boldsymbol{\mu}} \right) \hat{\boldsymbol{\mu}}, \lambda_{\min}(\Sigma) \right\}. \tag{125}$$

To complete the proof we should find a lower bound for the first term in the right-hand side of the above inequality. To this aim we write $\hat{\boldsymbol{\mu}}$ as follows:

$$\hat{\boldsymbol{\mu}} = \sum_{i=1}^d a_i \boldsymbol{u}_i, \tag{126}$$

where $\boldsymbol{u}_i$ is the $i$th eigenvector of $\Sigma$, and $\lambda_i$ is its $i$th eigenvalue, such that $\lambda_1 \leq \lambda_2 \leq \cdots \leq \lambda_d$. With this new description for $\hat{\boldsymbol{\mu}}$ we can write:

$$\hat{\boldsymbol{\mu}}^T \left( \Sigma - \frac{\boldsymbol{\mu}\boldsymbol{\mu}^T}{\boldsymbol{\mu}^T \Sigma^{-1} \boldsymbol{\mu}} \right) \hat{\boldsymbol{\mu}} = \sum_{i=1}^d a_i^2 \lambda_i - \frac{1}{\sum_{i=1}^d \frac{a_i^2}{\lambda_i}}$$

$$= \sum_{i=1}^d a_i^2 \lambda_i - \frac{\prod_{i=1}^d \lambda_i}{\sum_{i=1}^d a_i^2 \prod_{j\neq i} \lambda_i}$$

$$= \frac{\sum_{i,j=1}^d a_i^2 a_j^2 \lambda_i \prod_{k\neq j} \lambda_k - \prod_{i=1}^d \lambda_i}{\sum_{i=1}^d a_i^2 \prod_{j\neq i} \lambda_i}$$

$$= \frac{\left( \sum_{i=1}^d a_i^4 \prod_{i=1}^d \lambda_i - \prod_{i=1}^d \lambda_i \right) + \sum_{i,j=1}^d a_i^2 a_j^2 \lambda_i \prod_{k\neq j} \lambda_k}{\sum_{i=1}^d a_i^2 \prod_{j\neq i} \lambda_i}$$

$$= \frac{\sum_{i,j=1}^d a_i^2 a_j^2 (\lambda_i - \lambda_j)^2 \prod_{k\neq i,j} \lambda_k}{2 \sum_{i=1}^d a_i^2 \prod_{j\neq i} \lambda_i}$$

$$\geq \frac{\lambda_{\min}\Delta^2}{d\lambda_{\max}}, \tag{127}$$

where in the last inequality $\Delta$ is defined as follows:

$$\Delta(\Sigma) = \min \{ \lambda_i - \lambda_j \}, \ \forall i, j : \lambda_i \neq \lambda_j. \tag{128}$$

An important point in the above inequalities is that, if $\hat{\boldsymbol{\mu}}$ be in the direction of an eigenvector of $\Sigma$, then $\hat{\boldsymbol{\mu}}^T \left( \Sigma - \frac{\boldsymbol{\mu}\boldsymbol{\mu}^T}{\boldsymbol{\mu}^T \Sigma^{-1} \boldsymbol{\mu}} \right) \hat{\boldsymbol{\mu}}$ become equal to zero but in that case $\psi_1$ should be 0. The reason is that, in this situation $\hat{\boldsymbol{\mu}}$ and $\Sigma_1^{-1} \boldsymbol{\mu}$ will be in the same direction and we know that $\hat{\boldsymbol{\mu}}$ is perpendicular to $\Sigma_1^{-1} \boldsymbol{\mu}$. therefore we have:

$$\hat{\boldsymbol{\mu}}^T \left( \Sigma - \frac{\boldsymbol{\mu}\boldsymbol{\mu}^T}{\boldsymbol{\mu}^T \Sigma^{-1} \boldsymbol{\mu}} \right) \hat{\boldsymbol{\mu}} \geq \min \left\{ \frac{\lambda_{\min}\Delta}{\lambda_{\max}}, \lambda_{\min} \right\}, \tag{129}$$

and the proof is complete. $\qquad \square$

## D    EXPERIMENTS DETAILS

As mentioned in the manuscript, two different experiments were designed and implemented to demonstrate the performance of the proposed method. The codes are written using the Python programming language and the Pytorch 2.0 machine learning framework. The details of each test are explained below.

### D.1    EXPERIMENT ON SIMULATED DATA

#### D.1.1    SIMULATED DATA

The data employed in this particular section comprises two distinct classes, namely positive and negative. These classes are generated by sampling from Gaussian distributions with 200 dimensions denoted as $N(\mu, \sigma^2 I)$ and $N(-\mu, \sigma^2 I)$, respectively. The parameter $\mu$ is initialized randomly such that $||\mu||_2 = 1$. Also, for unlabeled out-of-distribution data, two different Gaussian distributions, $N(\mu', \sigma^2 I)$ and $N(-\mu', \sigma^2 I)$, respectively, have been considered for positive and negative classes, which $\mu' = \mu + \alpha.v$ where $\alpha = \frac{||\mu||_2}{k}, k \in \mathbb{N}$ and $v$ is a random normalized vector.

For the training and testing data, two classes, positive and negative, have been uniformly sampled in labeled and unlabeled modes. All the test data are taken from the main distribution ($N(\mu, \sigma^2 I)$ and $N(-\mu, \sigma^2 I)$) and it consists of 10,000 samples, half of which belong to the positive class and the other half belong to the negative class.

#### D.1.2    MODELING

A linear model has been used to learn these data, the purpose of which is to obtain optimal values for the weight vector w. The cost function for when labeled data is used is according to equation 130, where the objective is to maximize the margin of the linear classifier.

$$l_{\text{labeled}}(x, y) = \sum_{i=0}^{n} \min\left(1, \max\left(0, 1 - \gamma.y_i.\left(w^T.x_i\right)\right)\right). \tag{130}$$

In equation 131, $x_i$s are the feature vectors and $y_i$s are their corresponding labels, $\gamma$ is a regularization parameter for robust learning, $w$ is the weight vector of the linear model, and $n$ is the number of labeled samples.

The cost function for unlabeled samples is defined according to equation 131, where, due to the lack of access to labels for these samples, we use the model's predictions as their labels. Here too, the goal is to maximize the classifier's margin.

$$l_{\text{unlabeled}}(x') = \sum_{j=0}^{m} \min\left(1, \max\left(0, 1 - \gamma'.\|w^T.x_j'\|^2\right)\right) \tag{131}$$

In the above equation $x_i'$s are the feature vector of unlabeled samples, $\gamma'$ is a regularization parameter for robust learning, and $m$ is the number of unlabeled samples.

Finally, according to the proposed method, the combination of previous loss functions is used as the loss function of the model, when we have both labeled and unlabeled samples. We define this new loss function as follows:

$$l_{\text{total}}(x, y, x') = l_{\text{labeled}}(x, y) + \lambda.l_{\text{unlabeled}}(x') \tag{132}$$

In all of our experiments we use Adam optimizer Kingma & Ba (2015) with regularization term (weight-decay). Considering that the hyper-parameters of the problem must have an optimal value for each scenario; A random search process has been performed to find the optimum $\gamma$, $\gamma'$, $\lambda$, and weight-decay. Finally, we select a combination of hyper-parameters that achieved the highest accuracy on a validation dataset, and we report the accuracy of our model, using these hyper-parameters, on the test samples. This process is repeated for each experiment.

The details of the hyper-parameter values of the experiments reported for simulated data in the manuscript are available in the attached file `Hyperparameter-Simulated.csv`.

## D.2 Experiment on real data

### D.2.1 Dataset

To evaluate the performance of the proposed method on real-world datasets, we selected a set of histopathology data. The experiment is divided into two parts: one with labeled and unlabeled data from the same distribution, and another with labeled and unlabeled data from different distributions. The **NCT-CRC-HE-100K** dataset is used as the main distribution, and the **PatchCamelyon** dataset as the out-of-distribution dataset. Here are some additional explanations about each of the datasets used in the experiment:

1. **NCT-CRC-HE-100K**: This dataset contains 100,000 non-overlapping 224×224 pixels patches that are extracted from 86 whole slide histopathology images of human colorectal cancer (CRC) and normal tissue. It contains 9 different classes: Adipose (ADI), background (BACK), debris (DEB), lymphocytes (LYM), mucus (MUC), smooth muscle (MUS), normal colon mucosa (NORM), cancer-associated stroma (STR), colorectal adenocarcinoma epithelium (TUM) Katherm et al. (2018).

2. **PatchCamelyon**: This dataset contains 327,680 patches with 96×96 pixels from whole slide histopathology images of lymph node sections. It contains 2 different classes that show the patch is a tumor or normal B. S. Veeling et al. (2018).

We conducted two distinct experiments to evaluate the impact of adding unlabeled data. The first experiment focused on assessing the accuracy improvement by incorporating unlabeled data sampled from the same distribution as the initial dataset. The second experiment aimed to evaluate the accuracy improvement when unlabeled data was sourced from a distribution different from the original dataset. For the experiment involving samples from the same distribution, we used the **NCT-CRC-HE-100K** dataset. On the other hand, for the experiment involving out-of-domain samples, we used the **PatchCamelyon** dataset.

In the first experiment, where both labeled and unlabeled data are from the same distribution (**NCT-CRC-HE-100K**), we evaluated the performance considering the 9 classes present in the dataset. However, in the second experiment using the **PatchCamelyon** dataset with only two labels, we needed to align the labels with the **NCT-CRC-HE-100K** dataset. For this purpose, we merged the cancer-associated stroma (STR) and colorectal adenocarcinoma epithelium (TUM) classes and labeled them as "tumor", while the remaining categories were labeled as "normal".

### D.2.2 Modeling

As mentioned in the manuscript, the processing pipeline involves feeding the images through a pre-trained ResNet model that has been trained on the ImageNet dataset Deng et al. (2009); He et al. (2016), and the output embedding is saved for each image. In these experiments, we use the implementation of Chen & Krishnan (2022) for ResNet50 to extract the embeddings. The stored embeddings are then fed to a deep neural network to perform classification. This deep network consists of four 2048 layers with LeakyReLU activation function and one layer at the end with the size of the number of classes (the input has a dimension of 1024). As shown in Algorithm 1; In order to obtain the adversarial perturbed inputs based on the original input, we update the initial value of the input a number of times in the direction of the gradient. Algorithm 2 shows the training flow that uses the labeled and unlabeled data. This process can be used based on whether we are in the labeled-only mode or whether we are in the labeled and unlabeled combination mode. In this way, if we are in the labeled-only mode, the parts related to unlabeled samples will not be done, and the loss function will only include $\phi_i$.

To determine the optimal values for the hyperparameters, namely the Learning rate, weight decay, $\lambda$, $\alpha$, $\gamma$, and $\gamma'$, we employ a random search technique across their respective parameter spaces. Through this approach, we systematically explore various combinations of these hyperparameters and identify the most effective values in each experiment. Table 3 presents the search space for each hyperparameter used in the experiments.

The details of the hyperparameter values of the experiments reported for histopathology data in the manuscript are available in the attached file `Hyperparameter-Histopathology.csv`.

---

**Algorithm 1** Finding the adversarial perturbed input for original input data based on gradient ascent

---

**function** ADVERSARIAL-PERTURB$(x, y, \gamma, \alpha, N_s)$
    $x' = x$
    **for** step = $1, ..., N_s$ **do**                          ▷ Gradient ascent loop
        $p = model(x')$
        $\phi = CE(p, y) - \gamma.||x' - x||_2^2$                   ▷ CE: Cross entropy loss
        $\alpha = \alpha/s$
        $x' = x' + \alpha\nabla_{x'}\phi$
    **end for**
    **return** $x'$
**end function**

---

---

**Algorithm 2** The training loop

---

**Require:** Number of epochs $N_{ep}$, Number of perturbation steps $N_s$, Set of hyper-parameter $\{\gamma, \gamma',$
    $\lambda$, Learning rate of perturbation $\alpha\}$
    $L = \{(x_0, y_0), (x_0, y_0), \ldots, (x_N, y_N)\}$               ▷ Labeled data with size of N
    $U = \{x'_0, x'_1, \ldots, x'_M\}$                      ▷ Unlabeled data with the size of M
    $k = 2$
    $L_b = $ the set of batches of $L$                          ▷ batch size = N/k
    $U_b = $ the set of batches of $U$                       ▷ batch size = M/k
    **for** epoch = $1, \ldots, N_{ep}$ **do**
        **for** $(x_i, y_i), x'_j$ in $L_b, U_b$ **do**
            $x_i^p = $ ADVERSERIAL-PERTURB$(x_i, y_i, \gamma, \alpha, step)$
            $y_j^l = model(x'_j)$
            $x_j'^p = $ ADVERSERIAL-PERTURB$(x'_j, y'_j, \gamma', \alpha, step)$
            $p_i = model(x_i^p)$
            $p_j = model(x_j'^p)$
            $\phi_i = CE(p_i, y_i) - \gamma.||x_i^p - x_i||_2^2$
            $\phi_j = CE(p_j, y'_j) - \gamma'.||x_j'^p - x_j||_2^2$
            $l = \phi_i + \lambda.\phi_j$
            backpropagate loss ($l$) with gradient decent to the deep net and update the weightes
        **end for**
    **end for**

---

Table 3: The hyperparameter search space that we test randomly.

| Hyperparameter | Search space ($i \in \mathbb{N}$) | Description |
|---|---|---|
| Learning rate | $\{10^i| -5 \leqslant i \leqslant -1\}$ | learning rate for adam optimizer |
| Weight decay | $\{10^i| -7 \leqslant i \leqslant -2\}$ | regularization term for adam optimizer |
| $\lambda$ | $\{10^i| -5 \leqslant i \leqslant 2\}$ | coefficient of the unlabeled term in loss function |
| $\alpha$ | $\{10^i| -5 \leqslant i \leqslant 1\}$ | learning rate for adversarial perturbation function |
| $\gamma$ | $\{10^i| -7 \leqslant i \leqslant 2\}$ | coefficient of norm term in labeled loss function |
| $\gamma'$ | $\{10^i| -7 \leqslant i \leqslant 2\}$ | coefficient of norm term in unlabeled loss function |

