# OpenReview forum: "Out-Of-Domain Unlabeled Data Improves Generalization"
_ICLR.cc/2024/Conference — ICLR 2024 spotlight_

### Official Review · Reviewer_w3Ke · 2023-10-20

**Soundness:** 2 fair
**Presentation:** 2 fair
**Contribution:** 1 poor
**Rating:** 6
**Confidence:** 3

**Summary:**

The paper proposes techniques for semi-supervised learning using distributional robust optimization and self-training. In addition, the methods proposed can utilize unlabeled samples that come from a different (but similar) underlying distribution. The paper also describes theoretical results that show generalization bounds in situations where the data follows Gaussian distributions.

**Strengths:**

The research topic is very relevant since the usage robust learning methods can facilitate the usage of unsupervised samples. In addition, the usage of unsupervised samples corresponding with a different distribution is also of interest and common in practice. Also, the development of theoretical guarantees in these settings is very relevant

**Weaknesses:**

The paper's contributions are not clear. Firstly, there are several methods for semi-supervision based on DRO [Najafi et al., 2019], [R1], [R2]. The methodological contributions in the submitted paper with respect to those works are unclear. The usage of unsupervised samples from a different distribution seems novel in this topic, but the distribution shift assumed in the paper seems too simplistic and straightforward to address by DRO methods (small change in the covariates' marginal). In addition, the authors should extend the experimental results, compare with existing methods for semi-supervision and use more common benchmark datasets, otherwise it is not possible to assess the relation with existing methods.

[R1] Jose Blanchet and Yang Kang. Semi-supervised learning based on distributionally robust optimization. In Data Analysis and Applications. 2020.

[R2] Charlie Frogner, Sebastian Claici, Edward Chien, and Justin Solomon. Incorporating unlabeled data into distributionally robust learning. Journal of Machine Learning Research. 2021.

**Questions:**

Utilizing bounds as those in (8) does not seem very meaningful for the case of linear classifiers for which a bound based on the norm of the parameters (Rademacher complexity) would be tighter. Is that the case?

After (6), “Hilbert space of constrained probability measures" is a typo?

The usage of the word self-supervision can be misleading since the methods proposed utilize techniques that obtain pseudo-labels for unsupervised samples that are usually referred to as "self-training."

---

> ### Author Response · Authors · 2023-11-11
> **Response to Reviewer w3Ke**
>
> We would like to thank the reviewer for reading our paper and giving us feedback. The reviewer has raised three primary concerns, to which we offer the following responses:
>
> - **There are several previous works in this area, and the current work should be positioned w.r.t. them**: In the forthcoming revised version, we will provide clarification on the identified issue. Specifically, we will contextualize our work in relation to Najafi et al. (2019), [R1] and [R2]. For now, let us briefly delve into these works and highlight their connections to our paper. Najafi et al. (2019) employs DRO in a semi-supervised setting; however, their methodology is fundamentally different from ours. Their approach involves the utilization of 'self-training' to assign soft/hard labels to unlabeled data, subsequently merging all portions of the dataset for model training via ERM. In contrast, our approach utilizes unlabeled data exclusively to constrain the set of classifiers, aiming to avoid crowded regions. In other words, we exclusively consider classifiers whose decision boundaries are distanced from the unlabeled data. DRO serves as a tool in our approach, though not necessarily as the primary objective. [R1] has already been discussed within Najafi et al.'s paper, where strong assumptions are made regarding the extent of adversarial shifts in distributions, placing it outside the precise scope of our work. [R2] shares similarities with Najafi et al. (2019); however, instead of assigning artificial labels to unlabeled samples, [R2] employs them to delimit the ambiguity set and enhance understanding of the marginals. Consequently, i) the consideration of out-of-domain samples is omitted, and ii) the focus on unlabeled data is directed towards acquiring insights into the marginals, as opposed to restricting the set of classifiers in our work.
>
> - **The distribution shift assumed in the paper seems too simplistic and straightforward to address by DRO methods (small change in the covariates' marginal)**: We believe that our assumptions regarding distribution shifts are maximally inclusive. Our singular assumption regarding the covariates' marginals is that they should reside within a known Wasserstein distance of the true distribution. It is crucial to emphasize that no assumptions about the 'labeling rule' have been made, since we will not have access to labels for the unsupervised data anyway. Conversely, establishing a bound on the distance (Wasserstein distance in this study) between the marginals of the true and shifted distributions is necessary (also mentioned by Reviewer Nric). Without such constraints, unlabeled data fails to provide meaningful utility. In other words, one could consistently generate unlabeled samples from some fixed and unrelated distribution for 'any' machine learning problem, irrespective of the inherent characteristics of the datasets under consideration. Additionally, it is noteworthy that numerous significant works in this domain have assumed a very similar constraint on distribution shifts, as exemplified by [P1], [P2], [P3], [P4].
>
> [P1] Deng, Z., Zhang, L., Ghorbani, A. and Zou, J., 2021, March. Improving adversarial robustness via unlabeled out-of-domain data. In International Conference on Artificial Intelligence and Statistics (pp. 2845-2853). PMLR.
>
> [P2] Kumar, A., Ma, T. and Liang, P., 2020, November. Understanding self-training for gradual domain adaptation. In International Conference on Machine Learning (pp. 5468-5479). PMLR.
>
> [P3] Lee, J. and Raginsky, M., 2018. Minimax statistical learning with wasserstein distances. Advances in Neural Information Processing Systems, 31.
>
> [P4] Sinha, A., Namkoong, H. and Duchi, J., 2018, February. Certifying Some Distributional Robustness with Principled Adversarial Training. In International Conference on Learning Representations.
>
> - **Authors should extend the experimental results, compare with existing methods for semi-supervision and use more common benchmark datasets**: In response to the reviewer's suggestion, we are committed to augmenting our experimental results in the revised version. It is important to emphasize, however, that our primary contribution lies in the realm of theory. The experiments serve the dual purpose of demonstrating the implementability of our method, ensuring its polynomial-time nature, and empirically illustrating the efficacy of leveraging unlabeled data, even from different domains, particularly when employing our proposed approach.

---

> > ### Author Response · Authors · 2023-11-11
> > **Part 2 (Questions Section)**
> >
> > **Questions**
> >
> > - **Utilizing bounds as those in (8) does not seem very meaningful for the case of linear classifiers for which a bound based on the norm of the parameters (Rademacher complexity) would be tighter. Is that the case?**: The bounds in (8) directly stem from the corresponding Rademacher complexity (as detailed in the proof of Theorem 4.1). The inclusion of the dimension ($d$), as opposed to the norm of the weight vector, is a consequence of our assumption of 'general linear classifiers.' Consequently, the norm of the weight vector is not constrained, in contrast to models like Ridge or Lasso regression. For these alternative methods, our bounds can be reformulated into different expressions, as suggested by the reviewer.
> >
> > - **After (6), “Hilbert space of constrained probability measures" is a typo?**: Thank you for bringing this to our attention. It appears to be a typo. What we intended to convey is the subset of the space of probability measures, corresponding to the specified constraints.
> >
> > - **The usage of the word self-supervision can be misleading since the methods proposed utilize techniques that obtain pseudo-labels for unsupervised samples that are usually referred to as "self-training."**: Thank you for highlighting this concern. We acknowledge the issue and will rectify it in the final version. Additionally, we will limit the use of the term 'self-supervision' in most instances.
> >
> > Considering the explanations provided above, we hope that many of the reviewers' concerns have been adequately addressed. In light of this, can you please reconsider your rating?

---

> > > ### Comment · Reviewer_w3Ke · 2023-11-17
> > >
> > > I thank the authors for the expanded explanations provided for the relationship with other DRO methods for semi-supervision. However, I still consider the methodological contribution of the paper is not enough for this conference.
> > >
> > > Multiple semi-supervision methods have been proposed using a DRO approach and self-training. It is not clear if the methods presented are superior to existing techniques or address some of their limitations. Notice also that "avoiding crowded areas of the input space" is the approach followed by multiple methods for semi-supervision, including classic works such as "entropy regularization" (Grandvalet & Bengio, 2004) and semi-supervised support vector machines [R1].
> > >
> > > The authors mention in their reply that the main contribution is in the realm of theory, but the results presented are for the specific case of "mixture of Gaussians with clustering assumption." Such distributional assumption is often too simplistic to represent data in practice. In addition, it is well-known that quite strong theoretical results for semi-supervision can be obtained under such assumption (see for instance [R2]).
> > >
> > > [R1] Bennett, K., & Demiriz, A. Semi-supervised support vector machines. Advances in Neural Information processing systems, 1998.
> > > [R2] P. Rigollet. Generalization error bounds in semi-supervised classification under the cluster assumption. Journal of Machine Learning Research. 2007.

---

> > > > ### Author Response · Authors · 2023-11-17
> > > > **Response to Reviewer w3Ke (Round 2)**
> > > >
> > > > We thank the reviewer for dedicating time to review our response and provide feedback. In addressing the concerns raised, we offer the following responses:
> > > >
> > > > - The reviewer has mentioned that there are several methods using DRO and self-training. Although we have addressed this question in the previous round, we find it useful to provide further explanation. Our work introduces a novel integration of DRO and Semi-Supervised Learning (SSL), capitalizing on out-of-domain unlabeled samples to enhance the generalization bound. To the best of our knowledge, this approach is unprecedented, with no prior work, including those referenced by the reviewer, exploring this combination. It is also essential to highlight that our utilization of self-training diverges from the conventional approach of training a classifier on labeled samples and assigning hard/soft-labels to unlabeled samples using that classifier. Therefore, our work does not completely fall into the category of ``self-training" methods.
> > > >
> > > > - Regarding "avoiding crowded areas of the input space," it's essential to clarify that this is not our novelty or contribution; rather, it is a fundamental assumption in semi-supervised learning methods with no particular assumption on the data generation model (such as ours), where classifiers tend to navigate regions with low density. Without such assumptions, unlabeled samples may prove to be ineffective. Our novelty lies in the incorporation of DRO and the utilization of out-of-domain samples to avoid crowded areas. Comparisons between our work and papers cited by the reviewer, such as Grandvalet and Bengio, 2004, and semi-supervised support vector machines [R1], may not be entirely equitable. These papers lack the same level of theoretical guarantees as our work and operate within fundamentally different scopes.
> > > >
> > > > - The reviewer also mentioned that the "mixture of Gaussian assumption" is deemed too simplistic. However, it is important to highlight that this assumption, even in a simpler format when the two **isotropic** Gaussians are **well-separated**, is the **sole** focus of many papers such as [P1][P2][P3], all of which are well-established and recognized works in this field. Therefore, to position and compare our work with those studies, we have no choice but to consider this assumption. Making a more general assumption might be deemed impractical and non-comparable to the results of the most recent and best works in the field.
> > > >
> > > > - The reviewer referenced a paper [R2] by Rigollet, suggesting that in cases where the underlying distribution adheres to the cluster assumption, there exist works with better sample complexity. With due respect to the reviewer, we respectfully disagree. While we acknowledge [R2] as a seminal work in the theoretical analysis of semi-supervised learning, it is crucial to note that in the most general format of the cluster assumption defined in their paper, their method requires exponentially many unlabeled samples, without information about the class of the underlying distribution—a scenario that aligns with our paper.
> > > >
> > > > [P1] Schmidt, L., Santurkar, S., Tsipras, D., Talwar, K. and Madry, A., 2018. Adversarially robust generalization requires more data. Advances in neural information processing systems, 31.
> > > >
> > > > [P2] Carmon, Y., Raghunathan, A., Schmidt, L., Duchi, J.C. and Liang, P.S., 2019. Unlabeled data improves adversarial robustness. Advances in neural information processing systems, 32.
> > > >
> > > > [P3] Alayrac, J.B., Uesato, J., Huang, P.S., Fawzi, A., Stanforth, R. and Kohli, P., 2019. Are labels required for improving adversarial robustness?. Advances in Neural Information Processing Systems, 32.

---

> > > > > ### Comment · Reviewer_w3Ke · 2023-11-17
> > > > >
> > > > > Thanks for the additional details. Just few clarifications about my comments:
> > > > >
> > > > > -with "it is well-known that quite strong theoretical results for semi-supervision can be obtained under such assumption" I didn't mean that the results in [R2] show better sample complexity than the presented manuscript; I guess the theoretical results in both papers are difficult to compare. I just wanted to express that multiple works have shown strong theoretical results for semi-supervision under the mixture of Gaussians assumption. If I am not mistaken, the submitted manuscript does not describe the contribution in that aspect or describe such previous work for semi-supervision.
> > > > > -as I mentioned in my review, I feel the main contribution in the paper is in terms of adversarial robustness and not so much in terms of semi-supervision. This also seems to be consistent with the authors' responses. If the expertise of other reviewers is in the field of adversarial robustness, they could assess better than me the contribution for that field. Nevertheless, if the paper's focus is on adversarial robustness, at least the paper's writing would need a thorough revision.

---

> ### Author Response · Authors · 2023-11-17
> **Response to Reviewer w3Ke (Remainder of Round 2)**
>
> We would like to thank the reviewer once again for their fast response. In the following, we present the remainder of our response from Round 2:
>
> We believe it is noteworthy to emphasize again that our work resides at the intersection of Distributionally Robust Optimization (DRO) and Semi-Supervised Learning (SSL) when unlabeled data are out-of-distribution. With all due respect, the reviewer did not mention any specific papers in this particular intersection. Since the reviewer is concerned about the level of contribution in our work, we find it useful to enumerate some of our findings here:
>
> - We present a non-asymptotic bound on the joint utilization of labeled and unlabeled samples for adversarially robust learning, as detailed in Theorem 4.1. This result stands as the first of its kind and extends the findings of [p2] and [p3]. While these previous studies concentrate on the efficacy of unlabeled samples when a single labeled sample is adequate for linear classification of a non-robust classifier, they do not offer insights into the requisite number of unlabeled samples when multiple labeled samples are involved, especially in scenarios where the underlying distribution exhibits limited separation between the two classes.
>
> - We introduce a non-asymptotic bound for the integration of labeled and unlabeled samples in semi-supervised learning, as articulated in Theorem 4.2. We posit that the outcomes of this theorem are unparalleled in the existing literature. To underscore the significance of our findings, consider the following example: contemplate the sample complexity of supervised learning for a linear classifier. In the realizable setting, where positive and negative samples can be completely separated by a hyperplane, the sample complexity is $\mathcal{O}(d/\epsilon)$. In the non-realizable setting, this sample complexity escalates to $\mathcal{O}(d/\epsilon^2)$. A pivotal question in learning theory revolves around how to approach the sample complexity of $\mathcal{O}(d/\epsilon)$ in the non-realizable setting. For instance, the insights provided by Namkong et al. in [P4] delve into this inquiry. Notably, even with the awareness that the underlying distribution is a Gaussian mixture, the optimal sample complexity, as per [P5], exceeds $\mathcal{O}(d/\epsilon^2)$. Our work's outcome demonstrates that in the scenario where the underlying distribution is a Gaussian mixture and we possess $m = \mathcal{O}(d/\epsilon)$ labeled samples, coupled with $n = \mathcal{O}\left(\frac{d^3}{m^2\epsilon^8}\right) = \mathcal{O}\left(\frac{d}{\epsilon^6}\right)$ unlabeled samples (without knowledge of the class to which the underlying distribution belongs), one can achieve an error rate lower than or equal to the case of having access to $\mathcal{O}(d/\epsilon^2)$ labeled samples.
>
> - We formalize the incorporation of out-of-domain unlabeled samples into the generalization bounds of both robust and non-robust classifiers. We contend that this represents a novel contribution to the field, with its closest counterpart being [P6]. Notably, [P6] addresses a scenario where the underlying distribution is a Gaussian mixture with well-separated Gaussian components, a condition that is not a prerequisite for our results.
>
> [P4] Namkoong, H. and Duchi, J.C., 2017. Variance-based regularization with convex objectives. Advances in neural information processing systems, 30.
>
> [P5] Ashtiani, H., Ben-David, S., Harvey, N., Liaw, C., Mehrabian, A. and Plan, Y., 2018. Nearly tight sample complexity bounds for learning mixtures of gaussians via sample compression schemes. Advances in Neural Information Processing Systems, 31.
>
> [P6] Deng, Z., Zhang, L., Ghorbani, A. and Zou, J., 2021, March. Improving adversarial robustness via unlabeled out-of-domain data. In International Conference on Artificial Intelligence and Statistics (pp. 2845-2853). PMLR.

---

> > ### Comment · Reviewer_w3Ke · 2023-11-17
> >
> > I appreciate the detailed comments provided by the authors regarding the paper's contribution with respect to the state-of-the-art in this last response. I would like also to note that such contribution is not adequately articulated in the submitted manuscript, and that was my main concern regarding the paper.
> >
> > I think this discussion would be more productive if the authors describe how they could improve the manuscript to more clearly describe the paper's contribution or position it more in the topic of adversarial robustness instead of semi-supervision (or both). I would consider to raise my score if the authors satisfactorily update the manuscript along the lines of the last response and this overall discussion. I believe the .pdf in openreview can be updated during this period, so it would be useful if the authors do that showing the changes made in a different color

---

> > > ### Author Response · Authors · 2023-11-18
> > > **PDF Update, and Summary of Changes**
> > >
> > > We express our gratitude for the reviewer's active engagement and thoughtful interaction. We also extend our thanks for affording us the opportunity to enhance our manuscript's quality in order to increase the score. In response to the reviewer's feedback, we have made the following updates to the PDF file:
> > >
> > > - Additional explanations have been incorporated into the first two paragraphs of Section 1.1 (Prior works) to comprehensively address our discussion on the paper's contribution and its position in the existing literature. Moreover, we have explicitly clarified that our primary focus in this work is on the robustness aspect, rather than aiming to advance the SSL paradigm in general.
> > >
> > > - A new section, Section 1.2 (Main contributions), has been introduced. This section predominantly encompasses the presentation of our contributions, as outlined in our previous response to the reviewer.
> > >
> > > - Various typos highlighted by the reviewer w3Ke and other reviewers have been rectified in the revised version.
> > >
> > > -----------------------------
> > >
> > > We hope the above changes would encourage the reviewer to reconsider their score.

---

> > > > ### Comment · Reviewer_w3Ke · 2023-11-19
> > > >
> > > > I appreciate the effort made by the authors to better describe the paper's contributions as well as the changes made in the manuscript. I will raise my score accordingly

---

> > > > > ### Author Response · Authors · 2023-11-19
> > > > > **Response to Reviewer w3Ke**
> > > > >
> > > > > Thank you for acknowledging the improvements in our manuscript. We appreciate your reconsideration and the effort you've put into the review.

---

### Official Review · Reviewer_Nric · 2023-10-30

**Soundness:** 3 good
**Presentation:** 3 good
**Contribution:** 3 good
**Rating:** 8
**Confidence:** 3

**Summary:**

The authors propose a robust learning approach that leverages slightly out-of-domain unlabeled observations to improve generalization performance. Both theoretical and empirical analysis of the method are provided. The approach leverages a robust learning objective and minimizes this objective on a combination of labeled data and pseudo-labeled out-of-domain data. This approach forces the decision boundary to avoid crowded areas of the input space.

**Strengths:**

The method pushes the classifier to avoid crowded areas, which is similar in spirit to large margin methods. Making use of unlabeled data seems to improve its ability to do this. The analysis provides a novel non-asymptotic learning bound for Gaussian Mixtures. The method also has well motivated controls for dialing in the bias-variance trade-off.

**Weaknesses:**

How does this method compare against large-margin based methods? What if one treats the "slightly out-of-distribution" data as in distribution? Some comparison with similar approaches is warranted.

"given" is misspelled in the abstract.

While I did not work through all the details, the "new set" of bounds appear to be based heavily on an upper bound of the Rademacher complexity. I feel this really ought to be stated in the abstract and main body of the paper. I don't think that comparing Rademacher complexity with VC dimension based bounds is really an advancement.

Why doesn't the amount of "slightly out of domain" enter into the bound $n>\Omega(\frac{m^2}{d})$? Certainly data from totally different domains wouldn't improve generalization, right?

**Questions:**

Can the authors add more details about the nature of their bounds in the main text to either confirm or soften claims of novel methodology?

---

> ### Author Response · Authors · 2023-11-12
> **Response to Reviewer Nric**
>
> We express our gratitude to the reviewer for dedicating time to read our paper and offering valuable feedback, as well as for providing a favorable score. In addressing the concerns raised by the reviewer, we present the following responses.
>
> ------------------
>
> Concerning the weaknesses:
>
> - **How does this method compare against large-margin based methods? What if one treats the "slightly out-of-distribution" data as in distribution? Some comparison with similar approaches is warranted.**: There are some similarities between our method and large-margin approaches in the sense that assuming a large margin in the distribution of data likely yields favorable bounds using our method. Regarding the second question, treating "slightly out-of-distribution" data as in distribution may introduce the distance between distributions ($\alpha$)in the upper bound. This effect intensifies with an increase in the number of unlabeled samples compared to labeled ones, potentially resulting in a loss of the upper bound. We appreciate the reviewer for raising these valuable questions, and in our revised version, we will provide a more detailed explanation as requested.
>
> - **"given" is misspelled in the abstract.**: Thank you for bringing this to our attention; we will correct it.
>
> - **While I did not work through all the details, the "new set" of bounds appear to be based heavily on an upper bound of the Rademacher complexity. I feel this really ought to be stated in the abstract and main body of the paper. I don't think that comparing Rademacher complexity with VC dimension based bounds is really an advancement.**: We have refrained from making assumptions about the set of linear classifiers in our hypothesis set, allowing the norms of weight vectors to remain unbounded. Consequently, our derivation of a generic upper bound for Rademacher complexity relies on VC dimension, leveraging Massart’s lemma and Sauer’s lemma. This connection is evident in the appearance of dimension $d$ in our bounds. Our results are therefore comparable to the VC dimension-based bounds highlighted by the reviewer. However, it is essential to note that our primary objective in establishing these upper bounds was to elucidate the impact of the number of unlabeled samples in the denominator. While our approach aligns with VC dimension-based bounds, should someone impose assumptions on the class of classifiers leading to a superior upper bound compared to VC dimension in the numerator, such enhancements would be reflected in our bounds as well.
>
> - **Why doesn't the amount of "slightly out of domain" enter into the bound $n > \Omega(\frac{m^2}{d})$? Certainly data from totally different domains wouldn't improve generalization, right?**: Due to the result of Theorem $4.2$,  the upper bound for the generalization error is $\mathcal{O}\left(\left(\frac{d\alpha}{m} + \frac{d}{m}\sqrt{\frac{d}{m+n}}\right)^{1/4}\right)$.Therefore, if we have unlabeled samples from a totally different domain, the parameter $\alpha$ will increase, worsening the upper bound. On the other hand if we have $\alpha \leq \frac{d}{m}$, the lower bound for the number of unlabeled samples to achieve a better generalization bound is $n > \Omega(\frac{m^2}{d(1-\frac{\alpha m}{d})^2})$.The bound mentioned by the reviewer in Corollary 4.4 assumes a negligible distance between distributions. Additional explanation on this matter will be provided in the final version.
>
> -----------------------
>
> Concerning the Questions:
>
> - **Can the authors add more details about the nature of their bounds in the main text to either confirm or soften claims of novel methodology?**: Of course, we will add adequate details about this matter.
>
> -------------------------
>
> We hope most of the reviewer's concerns have been addressed through the above explanations.

---

> > ### Comment · Reviewer_Nric · 2023-11-22
> > **Author responses**
> >
> > The authors have addressed my concerns.

---

### Official Review · Reviewer_v7JJ · 2023-11-01

**Soundness:** 3 good
**Presentation:** 3 good
**Contribution:** 3 good
**Rating:** 8
**Confidence:** 3

**Summary:**

The advantage of unlabeled data in coming up with sample-efficient robust classifiers is well known and most studied in the canonical case of classifying a mixture of Gaussian models. Prior works that analyze the tradeoff between labeled and unlabeled data for robust classifiers consider robustness w.r.t. perturbing inputs, in other words, adversarial robustness; while this paper views robustness as obtaining unlabeled data from a perturbed distribution in a Wasserstein ball of some radius, commonly known as being 'distributionally robust'. The authors propose an algorithm that adds a regularizer to the conventional ERM loss function and obtain sample complexity bounds for this algorithm to obtain PAC-optimal linear classifiers for classifying a mixture of Gaussian models. The regularization term is derived from the computing a 'robust loss' on the unlabeled data using labels obtained from a model that is trained on the labeled data. The robust loss is taken from the dual form of the distributionally robust optimization problem, Experimental results on both real and simulated data show that using a large amount of unlabeled data almost achieves the performance of the optimal classifier.

**Strengths:**

While multiple works have proposed algorithms that show the advantage of unlabeled data for obtaining robust classifiers with high accuracy, this paper's contribution lies in providing an algorithm for the distributionally robust framework that has theoretical guarantees for the linear classification for the Gaussian mixture model case. The latter has also been studied in other robustness frameworks thus highlighting the significance of studying it in a different robustness setting. The algorithm seems quite natural since the robust loss has been considered in prior works. While I haven't thoroughly verified the correctness of the proofs, the theorems seem reasonable. The paper is well-written and clear barring a few exceptions.

**Weaknesses:**

A few correctable weaknesses follow that could help strengthen the paper:
1) The comparison with related works isn't thorough in the sense that the paper mentions these related works but doesn't provide any comparison of their current work with it. For e.g. the paper doesn't mention that the works of Carlini et al, Carmon et. al. etc. were for the adversarial robustness setting. It would make the contributions stand out more clearly if there is precise comparison as to how earlier work is different from the current.
2) The experimental results could be more systematic and require more explanation. Please see the following sections for specific questions that can help strengthen this part.
3) A few minor clarifications are required in the theoretical section. I have elaborated in the following section.

**Questions:**

1) There are multiple anomalies that are unexplained in the experiments. Perhaps addressing that would make the results less surprising. Few examples: i) In the simulated data, it would helpful to know roughly the least number of labeled samples required to obtain the same accuracy as the optimal. ii) In the real dataset case, the different distribution case achieves higher accuracy for fewer labeled samples. It is not clear why this column shouldn't be the same or worse as the same distribution case since it is using only the labeled samples that are drawn both from the NCT-CRC-HE-100K dataset?
2) It should be possible to empirically verify the dimension dependence?
3) On Pg5, what is "crowded areas"? This notion makes an appearance in the conclusion as well, but there is no explanation about what this is?
4) In Theorem 4.1, why is the expectation only w.r.t P_0 since we also get unlabeled samples from P_1. If this is included in the high probability statement, then that seems strange because I would assume the algorithm randomness to be separately considered from the unlabeled dataset randomness.
5) Is there any intuition for why the error in robust loss increases with \gamma?
6) Minor comment: I found the notation in (1), \hat{R}(\theta, S) confusing and also not used later (unless in the proofs). Isn't this just R(\theta, \hat{P}^{m}_S) ?

---

> ### Author Response · Authors · 2023-11-12
> **Response to Reviewer v7JJ**
>
> We would like to thank the reviewer for taking the time to read our paper and providing valuable feedback. Regarding the concerns raised by the reviewer, the following responses are suggested:
>
> ---------------------
>
> Concerning the weaknesses:
>
> - **The comparison with related works isn't thorough**: In the final version, we will incorporate additional explanations and provide a more detailed comparison with existing works to ensure that our contribution is distinctly highlighted.
>
> ------------------------
>
> Concerning the Questions:
>
> - **There are multiple anomalies that are unexplained in the experiments**: We appreciate the reviewer for highlighting this issue. In our final version, we will conduct additional experiments and provide a detailed discussion of these anomalies. Regarding the two examples mentioned by the reviewer: i) We will include these values in our revised version. ii) The purpose of the table was not to compare the two settings where unlabeled samples came from the same or different distributions, but rather to illustrate the impact of using more unlabeled samples in each case. Therefore, we used different test sets and search spaces for hyperparameter search in these two settings, resulting in different accuracies in these two parts. However, we recognize the need for clarity and will regenerate the table, accompanied by an explanation comparing these two settings in our final version.
>
> - **It should be possible to empirically verify the dimension dependence?**: Yes, we will add some experiments regarding the relationship between the number of samples and dimension in our final version.
>
> - **What is "crowded areas"?**: By "crowded areas," we are referring to locations where the probability density function exhibits higher values. Our constraints, based on Distributionally Robust Optimization (DRO) and relying solely on unlabeled data, inherently force the classifier boundaries to maintain a distance from these identified areas.
>
> - **In Theorem $4.1$, why is the expectation only w.r.t $P_0$ since we also get unlabeled samples from $P_1$. If this is included in the high probability statement, then that seems strange because I would assume the algorithm randomness to be separately considered from the unlabeled dataset randomness.**: The expectation in Theorem 4.1 is solely with respect to $P_0$, as these terms constitute the definition of generalization risk within the distribution for which we aim to develop a proficient classifier. Although we utilize labeled samples from $P_0$ and unlabeled samples from $P_1$ to construct a robust classifier, the stochastic nature associated with these samples is reflected in the upper bound provided for the Rademacher complexity and the high probability statement.
>
> - **Is there any intuition for why the error in robust loss increases with $\gamma$?**: The parameter $\gamma$ exhibits an inverse relationship with the robustness of the classifier, indicating that a lower value of $\gamma$ leads to a more robust classifier with reduced risk. Additionally, it is a well-established fact that a more robust classifier tends to have lower variance compared to a less robust one. Consequently, a higher value of $\gamma$ corresponds to increased variance and a less favorable upper bound.
>
> - **Minor comment: I found the notation in (1), $\hat{R}(\theta, S)$ confusing and also not used later (unless in the proofs). Isn't this just $R(\theta, \hat{P}^{m}_S)$ ?**: The reviewer's observation is accurate. We will rectify this issue to eliminate any potential confusion.
>
> -------------------------
>
> We hope most of the reviewer's concerns have been addressed through the above explanations.

---

> > ### Comment · Reviewer_v7JJ · 2023-11-20
> > **Response**
> >
> > Thanks for the clarifications and for the additions. I have updated my score accordingly.

---

> > > ### Author Response · Authors · 2023-11-20
> > >
> > > Thank you for acknowledging the improvements in our manuscript. We appreciate your reconsideration and the effort you've put into the review.

---

### Official Review · Reviewer_TZRA · 2023-11-01

**Soundness:** 3 good
**Presentation:** 3 good
**Contribution:** 2 fair
**Rating:** 6
**Confidence:** 4

**Summary:**

The paper aims to utilize unlabeled samples from a perturbed distribution to improve the generalization error in both adversarially robust and non-robust settings. It introduces a new algorithm that leverages adversarially robust optimization and self-supervised learning. Subsequently, by focusing on the linear Gaussian mixture model, the paper shows that the use of unlabeled samples can lead to an improvement in error rates compared to traditional ERM, which does not make use of unlabeled samples.

**Strengths:**

- The paper analyzes the generalization error of the newly introduced algorithm, which utilizes self-supervised learning and adversarially robust optimization, demonstrating an improvement in error compared to traditional ERM.

- It provides experimental results corroborating their theoretical findings that unlabeled samples from a perturbed distribution can reduce the test error.

**Weaknesses:**

- The paper's linear Gaussian mixture model is very restrictive.

- The manuscript dedicates a substantial portion to discussing established definitions and findings in the literature. In contrast, the final three pages primarily center on discussing the paper's contributions.

**Questions:**

When n=0, no out-of-domain samples are utilized and the problem reduces to simple ERM. But in Theorem 4.2, when n=0, the dependence of the error on dimension is d^{3/8}, meaning that this reduction in the exponent of the dimension is not related to the utilization of out-of-domain samples. Furthermore, in the case of n=0, why is the non-robust error better than error obtained through ERM?

---

> ### Author Response · Authors · 2023-11-11
> **Response to Reviewer TZRA**
>
> We would like to thank the reviewer for taking the time to read our paper and providing valuable feedback. Regarding the concerns raised by the reviewer, the following responses are suggested:
>
> ------------
>
> Concerning the weaknesses:
>
> - **The paper's linear Gaussian mixture model is very restrictive.**: It is crucial to emphasize that the algorithm and methodology outlined in the paper do not rely on specific assumptions about the distributions of the data, and consequently are NOT confined to a Gaussian mixture model. Our approach is inherently general. While conducting a theoretical analysis in the paper, we chose to assess the algorithm's performance within the framework of a Gaussian mixture model for two primary reasons: i) this assumption is prevalent in the existing literature [p1][p2][p3], thereby facilitating comparisons between our theoretical advancements and prior findings (a point also underscored by Reviewer v7JJ). Additionally, ii) employing a Gaussian mixture model allows us to derive mathematically explicit bounds, enabling us to elucidate the potentially surprising aspects of our findings.
>
> [p1] Schmidt, L., Santurkar, S., Tsipras, D., Talwar, K. and Madry, A., 2018. Adversarially robust generalization requires more data. Advances in neural information processing systems, 31.
>
> [p2] Carmon, Y., Raghunathan, A., Schmidt, L., Duchi, J.C. and Liang, P.S., 2019. Unlabeled data improves adversarial robustness. Advances in neural information processing systems, 32.
>
> [p3] Alayrac, J.B., Uesato, J., Huang, P.S., Fawzi, A., Stanforth, R. and Kohli, P., 2019. Are labels required for improving adversarial robustness?. Advances in Neural Information Processing Systems, 32.
>
> -------------------
>
> Regarding the questions raised:
>
> - **When $n=0$, no out-of-domain samples are utilized, and the problem reduces to simple ERM. But in Theorem $4.2$, when $n=0$, the dependence of the error on the dimension is $d^{3/8}$, meaning that this reduction in the exponent of the dimension is not related to the utilization of out-of-domain samples. Furthermore, in the case of n=0, why is the non-robust error better than the error obtained through ERM?**: Let us clarify this issue. When we do not use any unlabeled samples (i.e., $n=0$), our bound becomes $\mathcal{O}\left((\frac{d}{m})^{3/8}\right)$, which is actually worse than the upper-bound of ERM ($^*$), $\mathcal{O}\left((\frac{d}{m})^{1/2}\right)$, and thus the ERM algorithm gives a better generalization bound. It's also important to highlight that these bounds are meaningful only when $\frac{d}{m} \leq 1$. Additionally, in the presence of unlabeled samples, there is no requirement for $m$ to be greater than $d$ in our method. The upper-bound then becomes $\mathcal{O}\left(\left(\frac{d}{m + n}\right)^{3/8}\right)$ , and it tends to zero as $n$ approaches infinity. We will further elucidate this matter in the final version to avoid any ambiguity. Specifically, we acknowledge that Remark 4.3 may have contributed to this misunderstanding. Therefore, we are committed to revising its statement in the final version to ensure clarity.
>
> (*) The performance gap between our method and ERM when $n=0$ can be effectively minimized to zero by intentionally diminishing the impact of the constraint (reducing $\lambda$ in our primary optimization criteria) when $n$ is small. However, it is important to note that the primary emphasis of this study is on extreme regimes, particularly when $n\gg m$. Consequently, we have chosen to omit these secondary derivations to enhance the overall readability of the paper.
>
> ------------------
>
> We hope most of the reviewer's concerns have been addressed through the above explanations.

---

### Public Comment · ~Hongxin_Wei1 · 2023-11-21
**Related works of exploring benefits of OOD data**

Dear authors,

I am very interested in this theoretical work and appreciate the high quality of the analysis. From my perspective, it would be better if the authors can provide a discussion for the literature of exploring benefits of OOD data, in the part of related works. There are several papers that are related to this topic (two are mine):

1. Lee, Saehyung, et al., "Removing Undesirable Feature Contributions Using Out-of-Distribution Data.", ICLR, 2021.

2. Hendrycks, Dan, Mantas Mazeika, and Thomas Dietterich. "Deep anomaly detection with outlier exposure." ICLR, 2019.

3. Wei, Hongxin, et al. "Open-set label noise can improve robustness against inherent label noise. NeurIPS 2021.

4. Wei, Hongxin et al. “Open-Sampling: Exploring Out-of-Distribution data for Re-balancing Long-tailed datasets.” ICML 2022.

5. Zhu, Fei et al. "OpenMix: Exploring Outlier Samples for Misclassification Detection" CVPR 2023.

In addition to these, there might be some other works from safe semi supervised learning. I hope this can help the authors build a more comprehensive intoduction to the related works.

Best Regards,
Hongxin

---

> ### Author Response · Authors · 2023-11-22
>
> Thank you, Hongxin, for expressing interest in our work.
> Should the paper get accepted, we will make sure to incorporate a more thorough revision of existing works, including those you mentioned, in the final version.

---

> > ### Public Comment · ~Hongxin_Wei1 · 2023-11-22
> >
> > Thank you for the response. Good luck!

---

### Meta-Review · Area_Chair_GTDu · 2023-12-19

**Metareview:**

The paper proposes a method, that incorporates Distributionally Robust Optimization (DRO) into self-supervised training, to use unlabeled data points during training. The authors provably show in the case of Gaussian mixtures a substantial improvement in the generalization error obtained by their proposed method compared to ERM (also, in the presence of out-of-distribution samples).

All the reviewers (including myself) are in favor of accepting the paper. I recommend that the authors incorporate all the great comments from the reviewers (e.g. expanding the part on the main contributions, systematic comparison with the prior work, etc -- see the updated reviews).

**Justification For Why Not Higher Score:**

The paper could be considered for a spotlight, but judging from the comments and discussions by the reviewer I would not recommend it for an oral.

**Justification For Why Not Lower Score:**

--

---

### Decision · Program_Chairs · 2024-01-16

Accept (spotlight)